# Effects of undercutting and sliding on calving: a global approach applied to Kronebreen, Svalbard

**Dorothée Vallot**[1], **Jan Åström**[2], **Thomas Zwinger**[2], **Rickard Pettersson**[1], **Alistair Everett**[3], **Douglas I. Benn**[4], **Adrian Luckman**[5,6], **Ward J. J. van Pelt**[1], **Faezeh Nick**[7], **and Jack Kohler**[3]

[1]Department of Earth Sciences, Uppsala University, Sweden
[2]CSC - IT Center for Science, Espoo, Finland
[3]Norwegian Polar Institute, Fram Centre, N-9296 Tromsø, Norway
[4]School of Geography and Sustainable Development, University of St Andrews, St Andrews, Scotland, UK
[5]Department of Geography, Swansea University, UK
[6]Department of Arctic Geophysics, UNIS, The University Center in Svalbard, Longyearbyen, Norway
[7]Arctic Geology Department, University Centre in Svalbard, Norway

*Correspondence to:* Dorothée Vallot

## Abstract.

In this paper, we study the effects of basal friction, sub-aqueous undercutting and glacier geometry on the calving process by combining six different models in an offline-coupled workflow: a continuum-mechanical ice flow model (Elmer/Ice), a climatic mass balance model, a simple sub-glacial hydrology model, a plume model, an undercutting model and a discrete particle model to investigate fracture dynamics (Helsinki Discrete Element Model, HiDEM). We demonstrate the feasibility of reproducing the observed calving retreat at the front of Kronebreen, a tidewater glacier in Svalbard, during a melt season by using the output from the first five models as input to HiDEM. Basal sliding and glacier motion are addressed using Elmer/Ice while calving is modelled by HiDEM. A hydrology model calculates subglacial drainage paths and indicates two main outlets with different discharges. Depending on the discharge, the plume model computes frontal melt rates, which are iteratively projected to the actual front of the glacier at subglacial discharge locations. This produces undercutting of different sizes, as melt is concentrated close to the surface for high discharge and is more diffuse for low discharge. By testing different configurations, we show that undercutting plays a key role in glacier retreat and is necessary to reproduce observed retreat in the vicinity of the discharge locations during the melting season. Calving rates are also influenced by basal friction, through its effects on near-terminus strain rates and ice velocity.

## 1 Introduction

Accelerated discharge of ice into the oceans from land ice is a major contributor to sea level rise, and constitutes the largest source of uncertainty in sea level predictions for the twenty-first century and beyond (Church et al., 2013). To a large degree, this uncertainty reflects the limited understanding of processes impacting calving from tidewater glaciers and ice shelves, and associated feedbacks with glacier dynamics. In particular, calving occurs by the propagation of fractures, which are not explicitly represented in the continuum models used to simulate ice flow and glacier evolution.

Recently, it has been suggested that ocean warming could play an important role in determining glacier calving rate and acceleration, by impacting submarine melt rates (Holland et al., 2008a; Luckman et al., 2015). Straneo and Heimbach (2013) proposed two mechanisms responsible for the increase of submarine melt rates at the ice-ocean interface in Greenland: a warmer and thicker layer of Atlantic water in the fjords and an increase in subglacial discharge mainly during summer and autumn. Buoyant meltwater plumes entrain warm ocean water (Jenkins, 2011) and are thought to enhance melt undercutting (Slater et al., 2015) at the ice cliff triggering collapse of the ice above. Luckman et al. (2015) investigated controls on seasonal variations in calving rates and showed that calving variations at Kronebreen, the glacier this study focuses on, are strongly correlated with sub-surface ocean temperature changes linked to melt under-

cutting of the calving front. However, direct measurements of oceanic properties, ice dynamics, frontal geometries and mean volumetric frontal ablation rates are still too scarce to quantify the relationship between ocean processes, subglacial discharge and ice dynamics and one must rely on modelling. Complex coupled process models can help to gain a better understanding of the physics taking place at tidewater glacier fronts.

In previous modelling work (Van der Veen, 2002; Benn et al., 2007; Amundson and Truffer, 2010; Nick et al., 2010; Cook et al., 2012; Krug et al., 2014, 2015), the dynamics of ice masses have been simulated using continuum models, in which the continuum space is discretised and include processes of mass and energy balance. In addition to the lack of process understanding, continuum models cannot explicitly model fracture, but must use simple parameterisations such as damage variables or phenomenological calving criteria. These problems can be circumvented using discrete particle models, which represent ice as assemblages of particles linked by breakable elastic bonds. Ice is considered as a granular material and each particle obeys Newton's equations of motion. Above a certain stress threshold, the bond is broken, which allows the ice to fracture. Åström et al. (2013, 2014) showed that complex crevasse patterns and calving processes observed in nature can be modelled using a particle model, the Helsinki Discrete Element Model (HiDEM). Bassis and Jacobs (2013) used a similar particle model and suggested that glacier geometry provides the first-order control on calving regime. However, the drawback of these models is that due to their high computer resource demand, they only can be applied to a few minutes of physical time.

A compromise should be found by coupling a continuum model, such as Elmer/Ice, to a discrete model, such as HiDEM, to successively describe the ice as a fluid and as a brittle solid. Sliding velocities and ice geometry calculated with the fluid dynamic model are used by the discrete particle model to compute a new calving front position. The effect of subglacial drainage mixing with the ocean during the melt season is taken into account by using a plume model that estimates melt rates at the front according to pro-glacial observed ocean temperatures, subglacial discharge derived from surface runoff and ice front height.

In this paper, we use both the capabilities of the continuum model Elmer/Ice and the discrete element model HiDEM. We harness the ability of HiDEM to model fracture and calving events, while retaining the long-term ice flow solutions of a continuum approach. The aim is to investigate the influence of basal sliding velocity, geometry and undercutting at the calving front on calving rate and location. We determine the undercutting with a high resolution plume model calculating melt rates from subglacial discharge. The simple hydrology model that calculates the subglacial discharge, is based on surface runoff that is assumed to be transferred directly to the bed and routed along the surface of calculated hydrological potential. We illustrate the approach using data from Kronebreen, a fast-flowing outlet glacier in western Spitsbergen, Svalbard (topography, meteorological and oceanographic data, as well as horizontal surface velocity and front positions from 2013) to assess the feasibility of modelling calving front retreat (rate and position).

## 2    Study area

Kronebreen is a tidewater glacier, that flows into Kongsfjorden in Svalbard, one of the fastest glaciers in the archipelago. The glacier front position undergoes seasonal oscillations, showing advance during the winter and spring followed by retreat in the summer and autumn. Since 2011, the summer retreat has outpaced the winter advance, with an overall net retreat of $\sim 2$ km between 2011 and 2015 after a relatively stable period since the 1990s (Schellenberger et al., 2015; Luckman et al., 2015; Köhler et al., 2016). Velocities at the front can reach $5 \, \mathrm{m \, d^{-1}}$ in the summer with large seasonal and annual variations associated with basal sliding velocity (Vallot et al., 2017). In 2013, averaged velocities close to the front ranged from 2.2 to $3.8 \, \mathrm{m \, d^{-1}}$ in the summer and fell to $2 \, \mathrm{m \, d^{-1}}$ directly after the melt season. In 2014, however, they stayed relatively high (around $4 \, \mathrm{m \, d^{-1}}$) throughout the summer and progressively fell to $3 \, \mathrm{m \, d^{-1}}$ in the winter.

Plumes of turbid meltwater, fed by subglacial discharge, are observed adjacent to the glacier terminus during the melt season (Trusel et al., 2010; Kehrl et al., 2011; Darlington, 2015; How et al., 2017). There are two main discharge points and the northern plume is generally more active than the southern one. Sediment-rich fresh meltwater discharge is thus mixing with saline fjord waters and can lead to a significant melt rate at the front of the glacier. Large variations of marine processes are typical for arctic fjords and Kongsfjorden experiences significant influx of warm water masses during the summer (Cottier et al., 2005) as shown by observations presented by Nahrgang et al. (2014) of ocean temperatures of Kongsfjorden from moored observatories in 2012–2013. From October to mid-November 2012, the whole water column temperature was warm (4–5 °C). Thereafter, the upper 100 m became colder and in January 2013, the whole water column temperature dropped to 1–3 °C. From March to May, it approached 0 °C and started to increase again in May (1–3 °C). In August, the temperature had reached 3–4 °C and the upper 100 m increased particularly to reach 5–6 °C towards the end of the season. Fjord bathymetry (Howe et al., 2003; Aliani et al., 2016) and bed topography under the glacier systems (Lindbäck et al., 2017) reveal a glacier terminus thickness of about 150 m at the discharge locations with 100 m of submerged column (see Fig. 1). Close to the subglacial discharge locations, a changing grounding-line fan of sediments has been observed (Trusel et al., 2010) potentially ensuring a pinning point if the glacier were to advance in the future. Luckman et al. (2015) showed that calving rates are strongly correlated with subsurface fjord temperatures, indi-

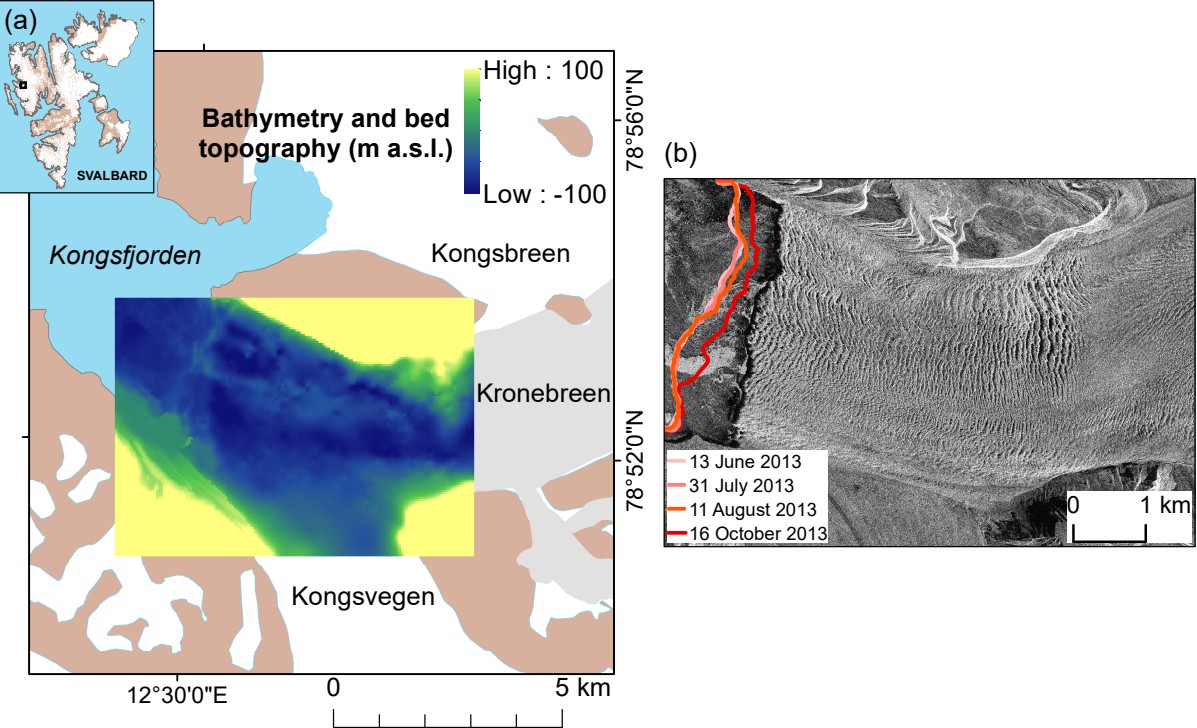

**Figure 1. (a)** Map of Kronebreen and its surrounding area. Ocean is in blue, bare rock is in brown and glacier ice is in white. The grey area represents the Kronebreen glacier system. The inset map top left shows the location of Kronebreen in Svalbard, and the central inset panel shows fjord bathymetry and bed topography in m a.s.l. **(b)** Crevasse pattern at the front of Kronebreen in August 2014 from TerraSAR-X satellite (1 m resolution), and four frontal positions during 2013.

cating that the dominant control on calving is melt undercutting, followed by collapse of the sub-aerial part.

## 3 Methods

### 3.1 Observed geometry, surface velocities and front positions

The bed topography, $z_b$, is derived from profiles of airborne and ground-based common-offset ice penetrating radar surveys distributed over Kronebreen from 2009, 2010 and 2014 (Lindbäck et al., 2017). The initial surface topography includes different available surface DEMs and is described in Vallot et al. (2017).

Ice surface velocities were derived from feature tracking of TerraSAR-X image pairs in slant range using correlation windows of $200 \times 200$ pixels at every 20 pixels, and subsequently ortho-rectified to a pixel size of 40 m using a digital elevation model (Luckman et al., 2015). Images were acquired roughly every 11 days for the period May–October 2013. Uncertainties in surface velocity are estimated to be $\sim 0.4\,\mathrm{m\,d^{-1}}$, and comprise a co-registration error ($\pm 0.2$ pixels) and errors arising from unavoidable smoothing of the velocity field over the feature-tracking window. Ice-front positions were manually digitised from the same im-

ages used for feature tracking after they had been orthorectified to a pixel size of 2 m using a surface DEM (Luckman et al., 2015).

### 3.2 Offline coupling approach

We use surface velocity and frontal position data described above to test the effects of sliding and undercutting on calving using different models in a global approach. This one-way offline coupling approach is divided into three parts using six models (see Fig. 2): inversion for sliding and computation of geometry evolution (with Elmer/Ice), determining undercutting (with the energy balance model, subglacial hydrology model, plume model and undercutting model) and computing calving (with HiDEM). In this paper, we use the output of five different models as input for the discrete particle model, HiDEM, in order to compare the modelled calving front to observations for different configurations of sliding, geometry and undercutting.

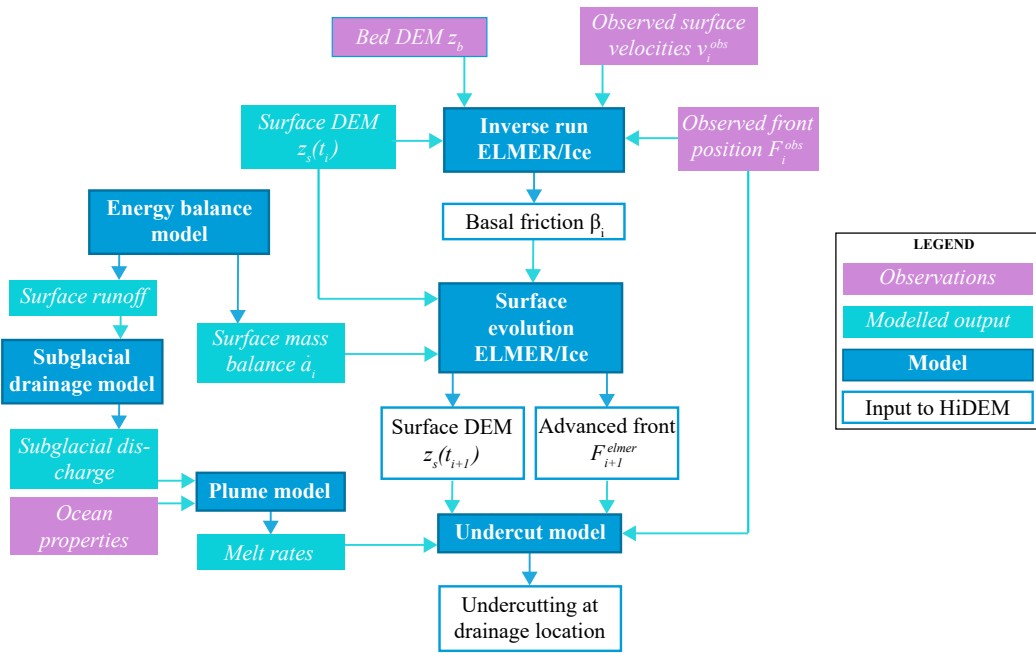

**Figure 2.** Model scheme presenting the calculation of the sliding and geometry (Elmer/Ice) as well as the undercutting at the subglacial discharge as input to the glacier calving from the HiDEM.

**Table 1.** Observation times of velocity acquisitions, $t_i$, associated dates and time interval between two observations ($\Delta t_i$). The HiDEM model is run for observational times $t_0$, $t_4$, $t_6$ and $t_{11}$ indicated by the gray color.

| $t_i$ | $\Delta t_i$ | Date | Comment |
|---|---|---|---|
| $t_0$ | | 2 June 2013 | Before the onset of the melting season |
| $t_1$ | 11 d | 13 June 2013 | First melt |
| $t_2$ | 11 d | 24 June 2013 | |
| $t_3$ | 11 d | 5 July 2013 | |
| $t_4$ | 26 d | 31 July 2013 | Period of high surface runoff |
| $t_5$ | 11 d | 11 Aug. 2013 | |
| $t_6$ | 11 d | 22 Aug. 2013 | Minimum basal friction |
| $t_7$ | 11 d | 2 Sept. 2013 | |
| $t_8$ | 11 d | 13 Sept. 2013 | |
| $t_9$ | 11 d | 24 Sept. 2013 | |
| $t_{10}$ | 11 d | 5 Oct. 2013 | |
| $t_{11}$ | 11 d | 16 Oct. 2013 | After the last melt |

We set $t_0$ at the velocity acquisition just before the first melt and the following observational times are set at each observation of surface velocity. The exact dates are summarized in Table 1.

First, we infer the sliding velocity at each observational time from surface velocities using an adjoint inverse method implemented in Elmer/Ice with an updated geometry from observations. At each iteration, $i$, corresponding to an observed front position, $F_i^{obs}$, the front and the surface are dynamically evolved during the observation time interval (roughly $11\,\mathrm{days}$) with Elmer/Ice with a time step of $1\,\mathrm{day}$. By the end of the observation interval, the front has advanced to a new position, $F_{i+1}^{elmer}$. Here we use $i+1$ because this is the position the front would have at $t_{i+1}$ in the absence of calving. Second, given subglacial drainage inferred from modelled surface runoff, a plume model calculates melt rates based on the subglacial discharge for each iteration, which are subsequently applied to the front geometry at subglacial discharge locations. At each iteration, the front geometry takes into account the undercutting modelled at the former iteration. Finally, the sliding velocity, geometry and undercutting (when applicable) are taken as input to the calving particle model HiDEM for each iteration and a new front, $F_{i+1}^{hidem}$, is computed for four iterations, $i = \{0, 4, 6, 11\}$, which represent interesting cases (see comments on Table 1). More details about each aspect of the model process are given in the following sections.

We call this approach an offline coupling because inputs to the HiDEM are output results from Elmer/Ice and undercutting model but not vice-versa. In Elmer/Ice, we use the observed frontal positions. A completely coupled physical model would use the output of HiDEM, the modelled front position, as input to the ice flow model Elmer/Ice and the undercutting model. It would also calculate the basal friction from a sliding law rather than an inversion. In principle, such

an implementation is possible using the same model components as this study.

### 3.3 Sliding and frontal advance with continuum model Elmer/Ice

5 At the base of the glacier, we use a linear relation for sliding of the form

$$\tau_{\mathbf{b}} + \beta \mathbf{v_b} = \mathbf{0}, \tag{1}$$

with $\tau_{\mathbf{b}}$, the basal shear stress and $\mathbf{v_b}$, the basal velocity. The basal friction coefficient, $\beta$, is optimized at each observa-10 tional time to best reproduce observed velocity distribution at the surface of the glacier as described in Vallot et al. (2017). This is done by using a self-adjoint algorithm of the Stokes equations for an inversion (e.g. Morlighem et al., 2010; Goldberg and Sergienko, 2011; Gillet-Chaulet et al., 2012) and 15 implemented in Elmer/Ice (Gagliardini et al., 2013). The inversion is performed using the method of Lagrange multipliers to minimise a cost function including the observed horizontal surface velocities and a Tikhonov regularisation. We use an unstructured mesh, with spatial repartition of el-20 ements based on the mean observed surface velocities in the horizontal plane (roughly $30\,\mathrm{m}$ resolution close to the front). Vertically, the 2D mesh is extruded with ten levels (roughly $10\,\mathrm{m}$ resolution close to the front). More details on the Elmer/Ice modelling (viscosity, ice temperature, iter-25 ations, etc.) are given in Vallot et al. (2017).

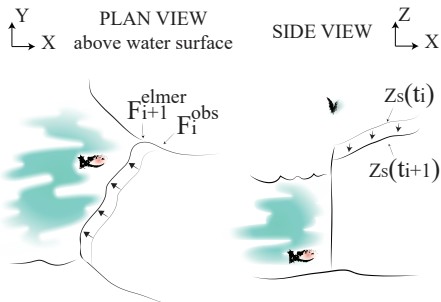

**Figure 3.** Front position and surface elevation changes with Elmer/Ice during $\Delta t = t_{i+1} - t_i$.

After each inversion, the temporal evolution of the glacier is mathematically described by the kinematic boundary condition defined at the surface,

$$\frac{\partial z_s}{\partial t} + v_x(z_s)\frac{\partial z_s}{\partial x} + v_y(z_s)\frac{\partial z_s}{\partial y} - v_z(z_s) = \dot{a}_s(t_i), \tag{2}$$

30 which describes the evolution of the free surface elevation, $z = z_s$, for a given net accumulation, $\dot{a}_s(t_i)$, calculated using a coupled modelling approach after Van Pelt and Kohler (2015), described in the next section. We use a time step of $1\,\mathrm{day}$ during the interval of time between two acquisitions. 35 Eq. 2 is solved alongside the Stokes equation, coupled to

the latter by the velocities. The basal sliding velocity is not evolved and stays equal to the result of the inversion. When the front is advanced, the mesh is stretched to match the new front position. No new element or node is created and the basal sliding coefficients are extrapolated towards the new 40 front. The new surface is in fact only used as an input for the next iteration. There is no interpolation of the basal sliding coefficients between two observational dates.

We assume that the front is vertical above the water line so that the observed front position (at the surface of the glacier) 45 is the same at sea level. We call $F_i^{obs}(z=0)$, the front position observed at time $t_i$ with $z=0$ at the sea level and $F_{i+1}^{elmer}(z=0)$, the advanced modelled front position after $\Delta t = t_{i+1} - t_i$ (see Fig. 3). The front is advanced by imposing a Lagrangian scheme over a distance equal to the ice ve-50 locity multiplied by the time step. We do not account for the submarine melting during the advance because we only have observations at the beginning and the end of each timespan. Instead, we lump frontal melting by applying an undercutting after the advance as explained hereafter. 55

### 3.4 Surface runoff and subglacial discharge model

The surface mass balance, $\dot{a}_s$, and runoff are simulated with a coupled energy balance-snow modelling approach (Van Pelt and Kohler, 2015). The coupled model solves the surface energy balance to estimate the surface temperature and melt 60 rates. The subsurface routine simulates density, temperature and water content changes in snow and firn while accounting for melt water percolation, refreezing and storage. The model is forced with 3-hourly meteorological time-series of temperature, precipitation, cloud cover and relative humidity from 65 the Ny-Ålesund weather station (eKlima.no; Norwegian Meteorological Institute). Elevation lapse rates for temperature are calculated using output from the Weather Research and Forecast (WRF) model (Claremar et al., 2012), while the precipitation lapse rate is taken from Van Pelt and Kohler 70 (2015); zero lapse rates are assumed for cloud cover and relative humidity. Surface runoff is modelled on a $100 \times 100\,\mathrm{m}^2$ grid.

The temporal subglacial discharge at the calving front is estimated from integration of daily surface runoff assumed 75 to be directly transferred down to the glacier bed. Assuming the basal water pressure at over burden, the flow path of the melt-water towards the glacier front is determined from the hydraulic potential surface defined as

$$\phi = \rho_i g(z_s - z_b) + \rho_w g z_b, \tag{3}$$

80

with $g$, the gravitational acceleration. The grid is the same as that used for surface runoff. The flow path along the hydraulic potential surface is determined by D-infinity flow method where the flow direction from a grid cell is defined as the steepest triangular facets created from the 8-neighboring 85 grid cells (Tarboton et al., 1987). The flow from the center grid cell is distributed proportionally to the two cells that de-

fine the steepest facet. The flow is accumulated as the melt water is routed along the calculated hydraulic potential surface towards the front and outlet points at the front are determined by identifying flow rates higher than $1\,\mathrm{m}^3\,\mathrm{s}^{-1}$. The hydraulic potential surface is filled before flow accumulation is calculated to avoid sinks.

### 3.5 Plume model and submarine melt rates

A high-resolution plume model is used here to simulate the behaviour of subglacial discharge at the terminus of Kronebreen. The model is based upon the fluid dynamics code Fluidity (Piggott et al., 2008) which solves the Navier-Stokes equations on a fully unstructured three-dimensional finite element mesh. The model formulation builds upon the work of Kimura et al. (2013), with the addition of a large eddy simulation (LES) turbulence model (Smagorinsky, 1963) and the use of the synthetic eddy method (SEM) at the inlet (Jarrin et al., 2006).

The geometry of the model is adapted to Kronebreen by setting the water depth to $100\,\mathrm{m}$ and initialising the model with ambient temperature and salinity profiles collected from ringed seals instrumented with GPS-equipped Conductivity, Temperature and Depth Satellite Relay Data Loggers (GPS-CTD-SRDLs) (Boehme et al., 2009; Everett et al., 2017). These data were collected between 14th August and 20th September 2012 from a region between one and five kilometers away from the glacier terminus and are taken as representative of the ambient conditions in the fjord during summer. Melt rates are calculated on the terminus using a three-equation melt parameterisation described by Jenkins and Bombosch (1995) and McPhee et al. (2008) and implemented in Fluidity by Kimura et al. (2013). Velocities driven by ocean circulation are typically around 2–3 orders of magnitude smaller than plume velocities and therefore neglected.

The model is spun-up for 1000 model seconds until the turbulent kinetic energy in the region of the plume reaches a steady state and thereafter run for 10 minutes of steady-state model time. Melt rates are extracted from the duration of the steady-state period, then time averaged and interpolated onto a uniform $1\times1\,\mathrm{m}^2$ grid covering a 400-m-wide section of the glacier terminus.

The high-computational cost of the model means that it cannot be run continuously over the study period, nor can the full range of discharges and oceanographic properties be tested. Instead, representative cases $M_d$ using the ambient ocean properties described above and discharges $d$ of 1, 10, 50 and $100\,\mathrm{m}^3\,\mathrm{s}^{-1}$ were tested and the melt rate profiles for intermediate discharges were linearly interpolated from these cases.

### 3.6 Undercutting model

We assume a vertically aligned surface front at the beginning of the melt season. We know the position of the front,

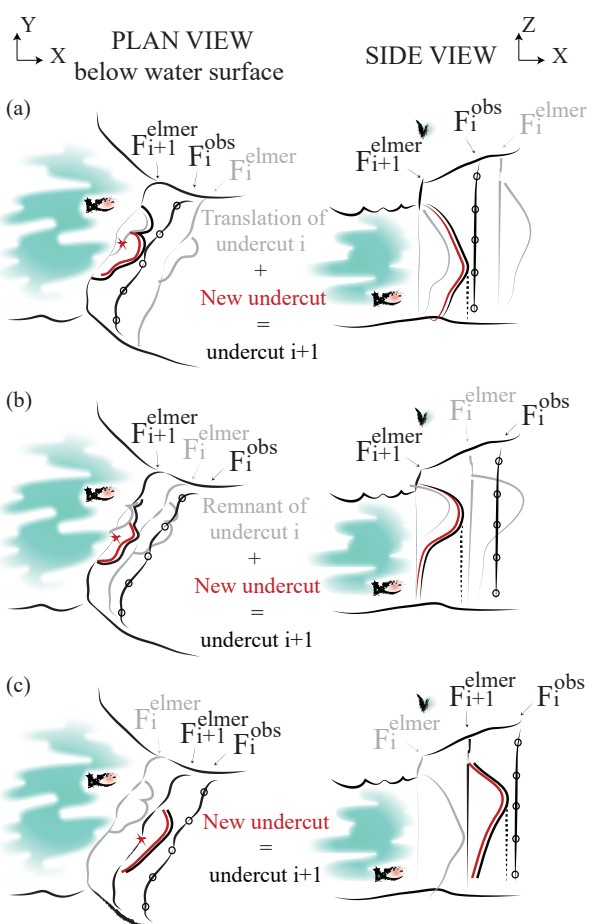

**Figure 4.** Three cases of undercutting $i+1$ at $t_{i+1}$ (black line) depending on former undercutting $i$ at $t_i$ (gray line) at $z$ relative to $F_i^{obs}(z=0)$ (black line with circles) in plan view (left) and side view (right). The red star represents the discharge location. On the side view, the dashed line represents the simplified undercut geometry where the ice foot has been removed, which is given as input to the HiDEM. **(a)** $F_i^{obs}(z=0)$ is behind $F_{i+1}^{elmer}(z=0)$ and in front of $F_i^{elmer}(z)$. The undercutting from $F_i^{elmer}(z)$ is translated to $F_{i+1}^{elmer}(z)$ (gray line) and the new undercutting is superposed (red line). **(b)** $F_i^{obs}(z=0)$ is in front of $F_i^{elmer}(z)$. The remnant from $F_i^{elmer}(z)$ (what is behind $F_i^{obs}(z=0)$) is translated to $F_{i+1}^{elmer}(z)$ (gray line) and the new undercutting is superposed (red line). **(c)** $F_i^{obs}(z=0)$ is behind $F_i^{elmer}(z)$. The undercutting from $F_i^{elmer}(z)$ is ignored and the undercutting created at $t_{i+1}$ is the only one (red line).

$F_0^{obs}(z=0)$, for the time span of each satellite image. The front is spatially digitized with $10\,\mathrm{m}$ spacing in the horizontal space and $1\,\mathrm{m}$ spacing in the vertical space. We use the combination of observed front, advanced front from Elmer/Ice and melt rates from the plume model to estimate the daily amount of undercutting. At each iteration, $i$, the sum of the daily undercutting during the observation interval is subtracted from the front.

When the first discharge occurs, the melt rate calculated with the plume model in 2D is summed for the period of time between $t_0$ and $t_1$ and projected to the advanced front $F_1^{elmer}(z=0)$ (advanced from $F_0^{obs}(z=0)$) at the location of the subglacial outlets and ice is removed normal to the front. This yields a new position of the front at depth $z$ below sea level called $F_1^{elmer}(z)$. At the second iteration, $t_2$, we know where the front would be if there had not been any calving between $t_1$ and $t_2$: $F_2^{elmer}(z=0)$, which is the advanced front from the observed position at $t_1$, $F_1^{obs}(z=0)$. So we can transfer the whole undercutting from previous iteration to $F_2^{elmer}(z)$ if $F_1^{obs}(z=0)$ is situated in front of $F_1^{elmer}(z)$ (see Fig. 4b–c). Otherwise, the undercutting would have been fully or partly calved away (see Fig. 4b–c). We then apply the new undercutting on this new geometry given the melt rates between $t_1$ and $t_2$.

At time $t_i$, the modelled front position at depth $z$ (advanced by Elmer/Ice from the observed front position at $t_{i-1}$) is $F_i^{elmer}(z)$ and the observed front position is $F_i^{obs}(z=0)$. We advance this observed front with Elmer/Ice during $\Delta t = t_{i+1} - t_i$ to obtain the front position $F_{i+1}^{elmer}(z=0)$ at $t_{i+1}$. We want to determine $F_{i+1}^{elmer}(z)$ and depth $z$ given the melt rate calculated between $t_i$ and $t_{i+1}$ and the state of the undercutting from the previous front $F_i^{elmer}(z)$ updated by the observed front $F_i^{obs}(z=0)$. Three different cases, depending on the relative position of the observed and modelled fronts at depth $z$, are then possible as shown in Fig. 4:

- if the new observed position $F_i^{obs}(z=0)$ is behind $F_i^{elmer}(z=0)$ and in front of $F_i^{elmer}(z)$, the melted undercutting is kept and advances in the flow direction the same distance as the surface modelled front $F_{i+1}^{elmer}(z=0)$ (see Fig. 4a),

- if the new observed position $F_i^{obs}(z=0)$ is in front of $F_i^{elmer}(z)$, the undercutting is displaced to the next modelled front $F_{i+1}^{elmer}(z=0)$ (see Fig. 4b),

- if the new observed position $F_i^{obs}(z=0)$ is behind $F_i^{elmer}(z)$, the front starts from a vertical profile again (see Fig. 4c).

The melt summed up between $t_i$ and $t_{i+1}$ is then applied to $F_i^{elmer}(z)$ to obtain $F_{i+1}^{elmer}(z)$ and so on. Frontal melt above the surface has not been taken into account so that the effect of submerged ice feet is not described. The bed topography, the new geometry (surface elevation, front position with or without undercutting) and the basal friction are then interpolated onto a $10 \times 10 \, \text{m}^2$ grid to feed the HiDEM and a new front, $F_{i+1}^{hidem}$ is modelled after calving for the four selected iterations ($i = \{0, 4, 6, 11\}$).

### 3.7 Calving with first-principles ice fracture model HiDEM

The fracture dynamics model is described in detail in Åström et al. (2013, 2014). This first-principles model is constructed by stacking blocks connected by elastic and breakable beams representing discrete volumes of ice. For computational efficiency, we use a block size of $10 \, \text{m}$.

At the beginning of a fracture simulation, the ice has no internal stresses and contains a few randomly distributed broken beams, representing small pre-existing cracks in the ice. The dynamics of the ice is computed using a discrete version of Newton's equation of motion, iteration of time steps and using inelastic potentials for the interactions of individual blocks and beams. As the ice deforms under its own weight, stresses on the beams increase, and if stress reaches a failure threshold the beam breaks and the ice blocks become disconnected but continue to interact as long as they are in contact. In this way cracks in the ice are formed. For computational reasons, we initialise the glacier using a dense packed face-centered cubic (fcc) lattice of spherical blocks of equal size. This introduces a weak directional bias in the elastic and fracture properties of the ice. The symmetry of the underlying fcc-lattice is however easily broken by the propagating cracks. The ground under the ice or at the sea-floor is assumed to be elastic with a linear friction law that varies spatially (Eq. 1).

The time step is limited by the travel time of sound waves through a single block and is thereby set to $10^{-4} \, \text{s}$. If the stress in the ice exceeds a fracture threshold, crevasses will form and ice may calve off the glacier. The duration of a typical calving event at Kronebreen is a few tens of seconds followed by a new semi-equilibrium when the ice comes to rest. The model run for $\sim 100 \, \text{s}$, which takes two days of computing time. As HiDEM cannot be triggered too often because of computational limitations, we simulate ice flow with Elmer/Ice and compute calving with HiDEM thereafter for the selected iterations. Calving events will then appear as fewer but bigger events compared to observations. If the time step is changed, the overall rate change stays roughly within $\pm 50\%$ (Benn et al., 2017).

The basal friction coefficients, $\beta$, at the front of Kronebreen are in the order of $10^8$-$10^{12} \, \text{kg m}^{-2} \, \text{s}^{-1}$ (Vallot et al., 2017) and to avoid instabilities to build up, a cut-off value, above which particles are assumed to be stuck to the bed substrate, is fixed at $\beta = 10^{12} \, \text{kg m}^{-2} \, \text{s}^{-1}$.

HiDEM reads a file with surface and bed coordinates on a grid and a file with surface and basal ice (to take into account the undercutting) coordinates. For simulations with an undercutting at a discharge location and in order to avoid complication in the HiDEM (position of the basal ice), we remove particles below the maximum melt (no ice foot as shown by the dashed line in Fig. 4). In the ocean, the basal friction coefficient is extrapolated downstream of the front and taken equal to the mean of the values further up from the terminus in case the ice advances. An ice block is calved when all bonds are broken from the glacier even though it does not separate from the front.

There is a clear separation of timescales between the velocities of sliding ($\sim \text{m day}^{-1}$) and calving ice ($\sim \text{m sec}^{-1}$).

This gives us the opportunity to rescale friction so that we can more effectively simulate calving: even if we scale down friction by e.g. two orders of magnitude and increase sliding accordingly to $\sim 100 \, \text{m day}^{-1}$, there is still negligible sliding during calving events which last tens of seconds or perhaps a minute. However, a rescaling speeds up the frequency of calving, and we can thus 'speed up', within reason, the few minutes of HiDEM simulation to effectively model calving which would otherwise take tens of hours or days, and thus be practically impossible to simulate with HiDEM. By applying scaling, the calving events modelled during the simulation of HiDEM (few minutes) correspond to the sum of calving events that would happen during the time scale of sliding. The scaling factor that we use is the same for the whole domain and for all simulations. We use a friction scaling factor for $\beta$ equal to $10^{-2}$ (or sliding velocity scaled up by $10^2$), and simulations run until calving stops and a new quasi-static equilibrium is reached.

In a fully coupled model, the altered ice geometry after calving could then be re-implemented in the flow model, acting as the initial state for a continued prognostic simulation with the continuum model. Here, this back-coupling is replaced by prescribing the next observed configuration.

### 3.8 Frontal ablation calculation

The mean volumetric frontal ablation rate (or mean volumetric frontal calving rate) at the ice front at time $t_i$, $\dot{a}_c(t_i)$, is the difference between the ice velocity at the front, $v_w(t_i)$ and the rate of change of the frontal position, $\partial L / \partial t$ integrated over the terminus domain $\Sigma_w$ as defined in McNabb et al. (2015). This yields

$$\dot{a}_c(t_i) = \int\limits_{\Sigma_w} v_w(t_i) - \frac{\partial L}{\partial t} \, \mathrm{d}\Sigma_w, \qquad (4)$$

with

$$\int\limits_{\Sigma_w} \frac{\partial L}{\partial t} \, \mathrm{d}A = \frac{\Delta A(t_i)}{t_i - t_{i-1}} \int\limits_{z \in \Sigma_w} \mathrm{d}z \qquad (5)$$

and $\Delta A(t_i)$, the area change at the terminus over the interval of time between two observations $t_i - t_{i-1}$. We want to compare the ablation rates from $F_i^{elmer}$ for observed and modelled cases. For the observed case, the mean volumetric ablation rate is calculated between the advanced front $F_i^{elmer}$ and the observed front $F_i^{obs}$. For the modelled case, during one simulation with HiDEM, several calving events are triggered. Volumetric calving rate is then inferred from the difference between the initial, $F_i^{elmer}$, and final position, $F_i^{hidem}$, of the front, after calving has stopped. The total subaqueous melt rate, $\dot{a}_m$, at the front of the glacier is omitted in this balance.

### 3.9 Calving scenario simulations

We investigate the effect of three different parameters on calving activity: the geometry, $g_i$, corresponding to the frontal position and topography, the sliding velocity mainly influenced by the basal friction parameter ($\beta_i$) and the undercutting, $u_i$, at the subglacial discharge locations for four distinct times $t_i = \{t_0, t_4, t_6, t_{11}\}$ (see Table 1). The different configurations are referred as $C(g_i, \beta_j, u_i)$. If $u_i = 0$, there is no undercutting, hence a vertical ice front at the subglacial discharge location. At $t = 0$, the melt season has not started yet so there is no modelled undercutting. At $t = 11$, the melt season is finished and there is no modelled undercutting. If $j \neq i$, the geometry, $g_i$, is taken at $t_i$ and the basal friction, $\beta_j$, at $t_j$ to assess the roles of geometry and basal sliding velocity. We investigate basal friction at $t_0$ and $t_6$ since the former has maximum friction and the latter minimum friction of the studied cases. The configurations studied in this paper are summarised in Table 2.

**Table 2.** Different configurations, $C$, characteristics and periods.

| Configuration | Characteristics | Applied to |
|---|---|---|
| $C(g_i, \beta_i, 0)$ | Geometry at $t_i$ <br> Sliding at $t_i$ <br> Vertical front | $i \in [0, 4, 6, 11]$ |
| $C(g_i, \beta_i, u_i)$ | Geometry at $t_i$ <br> Sliding at $t_i$ <br> Undercutting at discharge | $i \in [4, 6]$ |
| $C(g_i, \beta_j, 0)$ | Geometry at $t_i$ <br> Sliding at $t_j$ <br> Vertical front | $(i, j) \in [(0, 6)$ <br> $, (6, 0)]$ |

## 4 Results

### 4.1 Basal friction coefficients

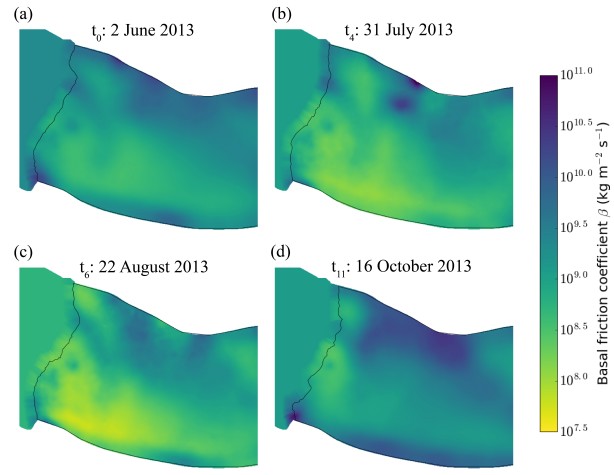

**Figure 5.** Basal friction coefficient obtained from inverse modelling and observed frontal position for **(a)** $t_0$: 2 June 2013, **(b)** $t_4$: 31 July 2013, **(c)** $t_6$: 22 August 2013 and , **(d)** $t_{11}$: 16 October 2013.

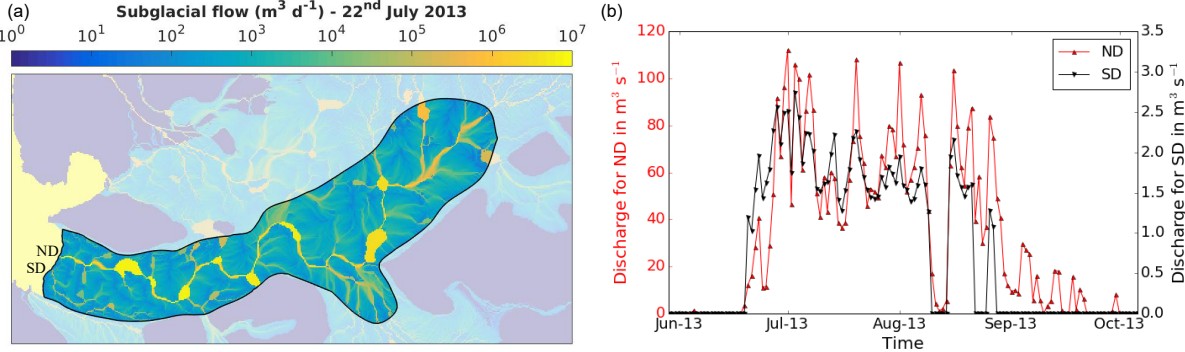

**Figure 6. (a)** Subglacial flow following the hydraulic potential surface (in $m^3\,d^{-1}$) in logarithmic scale on the 22nd July 2013. **(b)** Daily discharge for the northern and southern discharge (ND and SD respectively) during the melting season (data gaps correspond to no discharge).

The basal friction coefficient, $\beta$, for the four runs presented above, is shown in Fig. 5. At $t_0$, before the melt season, the basal friction is high and roughly homogeneous over the first kilometer. At $t_4$, when the surface runoff is the highest, the pattern is similar but with a large offset. The lowest friction is reached at $t_6$, particularly at the front and in the southern part of the glacier. The highest friction is reached at $t_{11}$ a kilometer from the front. Close to the front position, however, the friction is still high.

## 4.2 Subglacial discharge and submarine melt rates

The hydrological model predicts that there are two main subglacial channels with discharge exceeding $1\,m^3\,s^{-1}$ of water (see Fig. 6a). This is in accordance with satellite and time-lapse camera images showing upwelling at these locations (Trusel et al., 2010; Kehrl et al., 2011; Darlington, 2015; How et al., 2017). Modelled surface melt and discharge at the northern outlet – in short Northern Discharge (ND) – starts June 6 and ends October 1 while the discharge at the southern outlet (SD) starts June 21 and ends August 22. Fluxes at ND clearly exceed those at SD as shown in Fig. 6b and Table 3.

**Table 3.** Total volume of subglacial discharge modelled per period of calving front recording.

| Start date | End date | Days | Volume ($m^3$) ND | SD |
|---|---|---|---|---|
| 2 Jun ($t_0$) | 13 Jun ($t_1$) | 11 | 1.27e5 | |
| 13 Jun ($t_1$) | 24 Jun ($t_2$) | 11 | 8.73e6 | 4.94e5 |
| 24 Jun ($t_2$) | 5 Jul ($t_3$) | 11 | 6.24e7 | 2.05e6 |
| 5 Jul ($t_3$) | 31 Jul ($t_4$) | 26 | 1.10e8 | 3.54e6 |
| 31 Jul ($t_4$) | 11 Aug ($t_5$) | 11 | 6.2e7 | 1.36e6 |
| 11 Aug ($t_5$) | 22 Aug ($t_6$) | 11 | 4.69e7 | 1.04e6 |
| 22 Aug ($t_6$) | 2 Sept ($t_7$) | 11 | 3.91e7 | 2.03e5 |
| 2 Sept ($t_7$) | 13 Sept ($t_8$) | 11 | 1.18e7 | 0 |
| 13 Sept ($t_8$) | 24 Sept ($t_9$) | 11 | 6.20e6 | 0 |
| 24 Sept ($t_9$) | 5 Oct ($t_{10}$) | 11 | 8.04e5 | 0 |
| 24 Sept ($t_10$) | 5 Oct ($t_{11}$) | 11 | 0 | 0 |

The melt rate profiles calculated by the plume model for four different volumes of subglacial discharge are shown in Fig. 7.

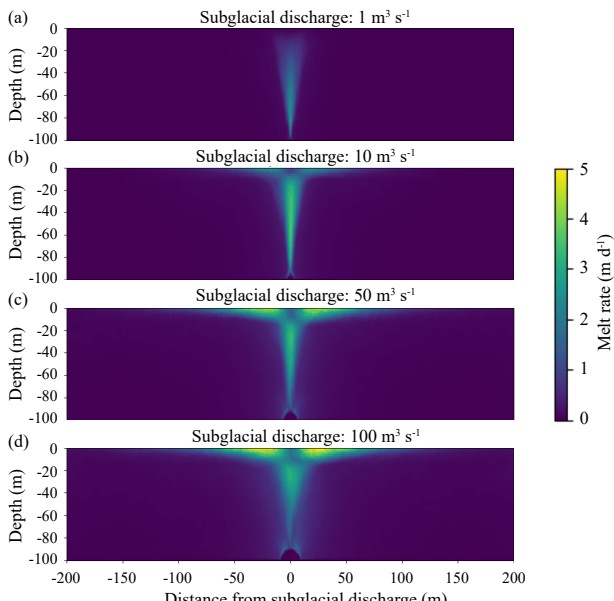

**Figure 7.** Melt rates, $M_d$, from the plume model given a discharge, $d$, of **(a)** $1\,m^3\,s^{-1}$, **(b)** $10\,m^3\,s^{-1}$, **(c)** $50\,m^3\,s^{-1}$ and **(d)** $100\,m^3\,s^{-1}$.

At a discharge of $1\,m^3\,s^{-1}$, melt rates are low ($< 2.5\,m\,d^{-1}$) with the maximum melt rate occurring at depth and negligible melt rates close to the water line. At $10\,m^3\,s^{-1}$, the melt profile reaches the surface and has highest melt rates ($\sim 3.5\,m\,d^{-1}$) along the plume column. With $50\,m^3\,s^{-1}$ and $100\,m^3\,s^{-1}$ discharge, the highest melt rates are attained at the ocean surface on the sides of the plume column ($\sim 5\,m\,d^{-1}$ and $\sim 6\,m\,d^{-1}$ respectively). In general, low discharges drive maximum melt within the plume and at depth, while higher discharges drive stronger surface gravity currents, and therefore gives higher melt rates at the surface.

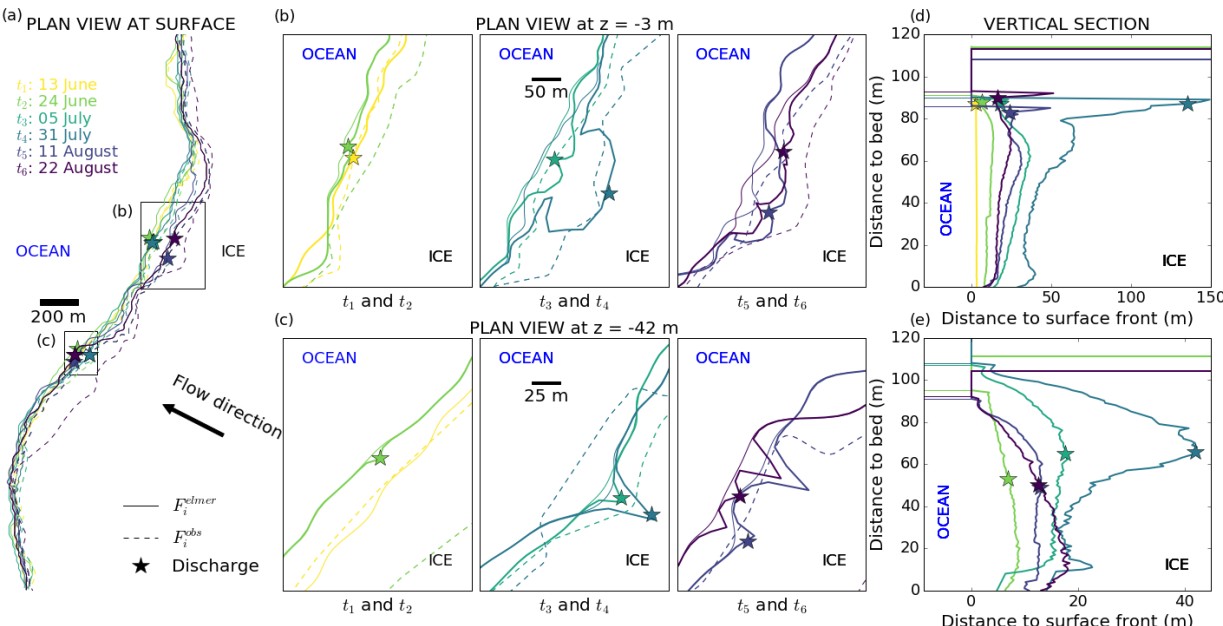

**Figure 8. (a)** Plan view of the observed frontal position of Kronebreen at six different dates, defined by different colors, corresponding to the satellite data acquisition dates during the melt season in 2013 (up to the 22nd of August). At $t_i$, the observed front, $F_i^{obs}$, is represented by a dashed line and the advanced front, $F_i^{elmer}(z=0)$, by a thin line. The discharge location is defined by a star. Enlargement at **(b)** the northern discharge (ND) area at $z = -3$ m and at **(c)** the southern discharge (SD) area at $z = -42$ m with the advanced front at depth $z$ where undercutting has been applied, $F_i^{elmer}(z)$, represented by a thick line. Vertical section **(d)** at the northern discharge (ND) location and at **(e)** the southern discharge (SD) location. The stars in (d,e) indicate the plan view elevation $z$ from (b,c). Horizontal lines in **(d, e)** represent the sea level for each iteration.

## 4.3 Undercutting

The modelled frontal position is summarised in Fig. 8 in plan view and vertical view at the discharge locations. In most cases for the ND location, where the discharge is the highest, the melt profile (Fig. 8d) creates an undercut profile concentrated right near the waterline. Fried et al. (2015) found similar results when modelling melt rates at shallow grounding lines (100–250 m) given 250 m$^3$ s$^{-1}$ discharge. It is interesting to see that the observed front after calving, $F_i^{obs}$, (dashed line in Fig. 8a–b) generally falls behind the undercut front before calving, $F_i^{elmer}(z)$, (thick line in Fig. 8b).

The frontal submerged undercutting driven by the plume differs in shape from one location to another. In the first 50 m below the surface, the undercutting at the SD is not as abrupt as at the ND and is also smaller (Fig. 8c–e). Where the discharge is the highest, the melt rate peaks just below the waterline and stretches laterally from the vertical centerline of the plume. The lateral extent of melting is much lower at depth. At the SD, melting is strongest at depth due to lower discharge rates and less vigorous buoyant ascent of the plume. One should keep in mind that our modelling approach neglects the change of the front during the period of interest between two observations of frontal positions (11 days for most cases). In reality, calving would occur more often during that period, making such large undercuttings, as the

modelled ones, not possible. This simplification has consequences for the next step when the particle model handles the calving of icebergs due to front imbalance.

## 4.4 Observed mean volumetric calving rates and modelled calving

The observed mean volumetric calving rate averaged over the entire calving front volume of ice, $\dot{a}_c^{obs}$ is the difference between the frontal velocity, $v_w^{obs}(t_i)$, and the rate of position change, $\partial L^{obs}/\partial t$ integrated over the terminus domain. These quantities and the total modelled ice mass melted by the plume normalised per day (when an undercutting is prescribed) are given in Table 4.

To assess the performance of the offline coupling, we evaluate the mean volumetric calving rate averaged over the entire calving front volume of ice (see Eq. 4), and the mean absolute distance between the modelled and the observed front, $|\bar{L}|$. These are presented in Fig. 9 for each configuration as well as the observed mean volumetric calving rate. Fig. 10 shows the different front positions after the HiDEM simulation for each configuration of the studied time. Fig. 11 shows strain rates modelled by HiDEM that resemble an observed crevasse patterns (yellow lines representing crevasses).

At $t_0$, before the melt started, the front has retreated at a rate of $7.93 \times 10^5$ m$^3$ d$^{-1}$ with a frontal ice flux of

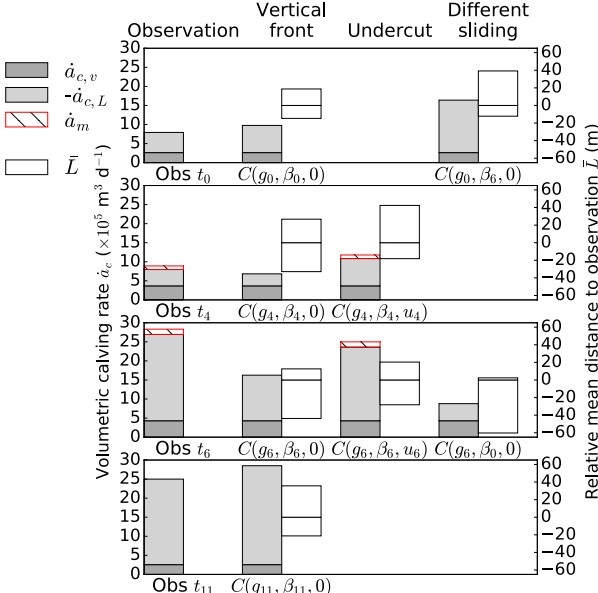

**Figure 9.** Observed and modelled mean volumetric calving rates, $\dot{a}_c$, in $\mathrm{m^3\,d^{-1}}$ are presented as the integrated tangential (ice flow direction) ice front velocity $\dot{a}_{c,v}$ (dark gray), the integrated rate of change of the frontal position, $\dot{a}_{c,L}$ (light gray) and the total subaqueous melt rate, $\dot{a}_m$ (red) if an undercutting is prescribed for each configuration. The mean distance differences between the modelled and the observed front positions, $\bar{L}$ are shown on the right. A negative value corresponds to underprediction of calving position (modelled in front of observed).

**Table 4.** Observed mean volumetric calving rate, $\dot{a}_c^{obs} = \dot{a}_{c,v}^{obs} - \dot{a}_{c,L}^{obs}$, in $10^5\,\mathrm{m^3\,d^{-1}}$, as the difference between the tangential (ice flow direction) ice velocity at the front and the rate of change of the frontal position integrated over the terminus domain, and estimated subaqueous melt rate, $\dot{a}_m$, in $10^5\,\mathrm{m^3\,d^{-1}}$.

|  |  | $t_0$ | $t_4$ | $t_6$ | $t_{11}$ |
|---|---|---|---|---|---|
| $\dot{a}_c^{obs}$ | $\dot{a}_{c,v}^{obs}$ | 2.63 | 3.68 | 4.31 | 2.56 |
|  | $\dot{a}_{c,L}^{obs}$ | −5.30 | −4.28 | −22.63 | −22.43 |
|  | **Total** | 7.93 | 7.97 | 26.94 | 24.99 |
| $\dot{\mathbf{a}}_\mathbf{m}$ | **SD** | 0 | 0.08 | 0.14 | 0 |
|  | **ND** | 0 | 0.86 | 1.25 | 0 |
|  | **Total** | 0 | 0.94 | 1.39 | 0 |
| *Ratio $\dot{\mathbf{a}}_\mathbf{m}/\dot{\mathbf{a}}_\mathbf{c}$* | | *0 %* | *11.8 %* | *5.2 %* | *0 %* |

$2.63 \times 10^5\,\mathrm{m^3\,d^{-1}}$, mostly in the middle part with a calved area of $5.1 \times 10^4\,\mathrm{m^2}$. The HiDEM produces a slightly higher mean volumetric calving rate, $9.76 \times 10^5\,\mathrm{m^3\,d^{-1}}$ with a vertical ice front configuration (red line $C(g_0, \beta_0, 0)$ in Fig. 10a) at a mean distance of $32\,\mathrm{m}$ from the observed front. However, calving is concentrated south of SD in a zone of high ice velocity and high strain rates as modelled by HiDEM (see Fig. 11).

With peak surface runoff, at $t_4$, the observed mean volumetric calving rate equals $7.97 \times 10^5\,\mathrm{m^3\,d^{-1}}$, similar to $t_0$

but with higher ice velocities ($3.68 \times 10^5\,\mathrm{m^3\,d^{-1}}$). Observed retreat at and north of ND is significant but is not reproduced by the configuration with a vertical ice front (red line $C(g_4, \beta_4, 0)$ in Fig. 10b). Instead the front is retreating south of SD in the same fashion as for $t_0$. The mean volumetric calving rate ($6.82 \times 10^5\,\mathrm{m^3\,d^{-1}}$) is therefore close to the observed value, but the mean distance between the observed and the modelled front is close to 60 m (see Fig. 9). For the undercutting configuration (blue line $C(g_4, \beta_4, u_4)$ in Fig. 10b), the mean volumetric calving rate is also overestimated at the same location but the observed retreat around ND is matched by the HiDEM. The mass removed by undercutting represents $11.8\,\%$ of the total observed mean volumetric calving rate (see Table 4) and is therefore non-negligible. At the SD, the observed front is advancing (see Fig. 8b) and regardless of the applied modelled front configuration (with or without undercutting), a similar slight retreat is modelled. In this case, the undercutting has no influence on the calving.

Vertical front configuration at $t_6$ (red line $C(g_6, \beta_6, 0)$ in Fig. 10c), during a period of accelerated glacier flow, results in slower modelled mean volumetric calving rate ($16.26 \times 10^5\,\mathrm{m^3\,d^{-1}}$) than observed ($26.94 \times 10^5\,\mathrm{m^3\,d^{-1}}$) and no front position change at both SD and ND leading to a mean distance to the observed front close to 60 m. With the undercut configuration (blue line $C(g_6, \beta_6, u_6)$ in Fig. 10b), modelled mean volumetric calving rate ($23.60 \times 10^5\,\mathrm{m^3\,d^{-1}}$) is similar to observation and the front positions at discharge locations are reproduced even though the undercutting only represents $5.2\,\%$ of the observed mean volumetric calving rate. The modelled front is still intensively breaking up south of SD but, at that date, it matches the observed retreat.

At the end of the melt season at $t_{11}$, when subglacial discharge has ended, the observed front retreats at a rate of $24.99 \times 10^5\,\mathrm{m^3\,d^{-1}}$ despite a frontal basal friction higher than at the last studied iteration resulting to an averaged frontal velocity of $2.56 \times 10^5\,\mathrm{m^3\,d^{-1}}$. But as shown in Fig. 5, the sliding velocity is higher (lower basal friction, $\beta_{11}$) close to the front than further upglacier. Large calving events occur at both former discharge locations where the bed elevation is lower than anywhere else. The calving front modelled by HiDEM (red line $C(g_{11}, \beta_{11}, 0)$ in Fig. 10d) manages to reproduce this behaviour but overestimates the retreat for the region in between, where the pattern of high strain rate is also denser (see Fig. 11).

Two configurations vary the friction coefficient, $\beta$, to assess the role of sliding in the calving process. If the basal friction is set according to $t_6$ and the geometry to $t_0$ (orange line $C(g_0, \beta_6, 0)$ in Fig. 10a), the mean volumetric calving rate exceeds observations by more than a factor of 2 ($16.40 \times 10^5\,\mathrm{m^3\,d^{-1}}$), similar to $C(g_6, \beta_6, 0)$, yet with matching spatial frontal patterns as $C(g_0, \beta_0, 0)$ as well as strain rate distribution with elevated rates close to the calved zones. If the geometry of $t_6$ is simulated with the basal friction of $t_0$ (orange line $C(g_6, \beta_0, 0)$ in Fig. 10c), it is striking

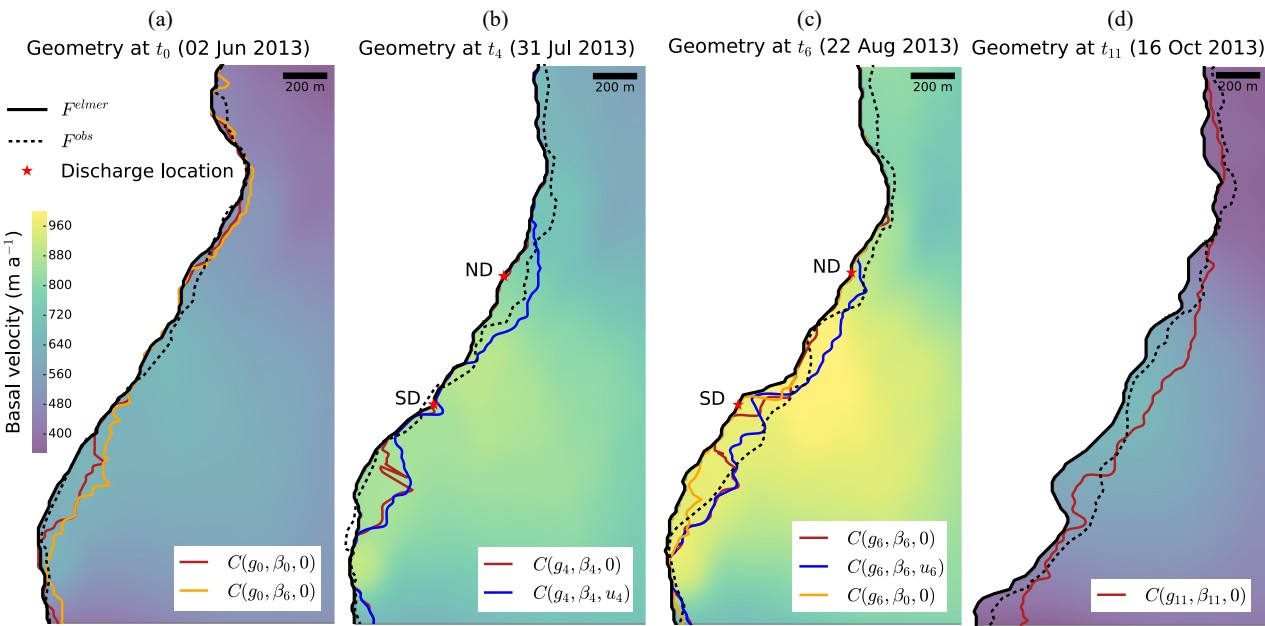

**Figure 10.** Basal velocity, advanced front before calving modelled with Elmer/Ice, $F_i^{elmer}$, at $t_i$ in plain black, observed front after calving, $F_i^{obs}$, in dashed black and calving front modelled with HiDEM, $F_i^{hidem}$ given the different configurations summarised in Table 2 for **(a)** $i = 0$, **(b)** $i = 4$, **(c)** $i = 6$, and **(d)** $i = 11$. Discharge locations (for $i = 4, 6$) are marked with a red star.

to notice again that the calved zones are similar to the vertical front configuration at $t_6$ but the mean volumetric calving rate is similar to the observed one at $t_0$. High strain rates are less pronounced than with the basal friction of $t_6$ but concentrated at the same locations.

## 5 Discussion

### 5.1 Plume Model and Undercutting

Our plume model uses a fixed, planar ice front to calculate submarine melt rates rather than a time-evolving geometry. This assumption is supported by Slater et al. (2017a), who showed that the shape of the submerged ice front does not have a significant feedback effect on plume dynamics or submarine melt rates. However, the same study suggests that the total ablation driven by submarine melting will increase due to the greater surface area available for melting. To take this effect into account in our undercutting model, submarine melt rates are horizontally projected onto the undercut front modelled at the previous iteration.

By using ambient temperature and salinity profiles that do not vary in time, we neglect the inter- and intra-annual variability in Kongsfjorden. This variability can affect the calculated melt rate in two ways: i) the three-equation melt parameterisation explicitly includes the temperature and salinity at the ice-face, and ii) the ambient stratification affects the vertical velocity and neutral buoyancy height of the plume. The direct effect of changes in temperature and salinity on

the melt equations are well tested. Past studies using uniform ambient temperature and salinity conditions have found a linear relationship between increases in ambient fjord temperatures and melt rates, with the slope of the relationship dependent upon the discharge volume (Holland et al., 2008b; Jenkins, 2011; Xu et al., 2013). Salinity, on the other hand, has been shown to have a negligible effect on melt rates (Holland et al., 2008a). However, with a non-uniform ambient temperature and salinity, the effects of changes in the stratification on the plume vertical velocity and neutral buoyancy are much more complex. The stratification in Kongsfjorden is a multi-layer system, with little or no direct relationship between changes in different layers (Cottier et al., 2005). Therefore, testing cases by uniformly increasing or decreasing the salinity would not be informative for understanding the true effects of inter- and intra-annual variability. The high-computational expense of the plume model used here means that it is not yet feasible to run the model on the timescales necessary to understand this variability, nor to run sufficient representative profiles to provide a useful understanding of the response. Previous work has suggested that intra-annual changes in the ambient stratification are small enough that plumes are relatively insensitive to these changes (Slater et al., 2017b) and that plume models forced with variations in runoff and a constant ambient stratification can qualitatively reproduce observations (Stevens et al., 2016). For these reasons, we highlight this as a limitation of the current implementation, and suggest that this should be addressed in future investigations of plume behaviour. A

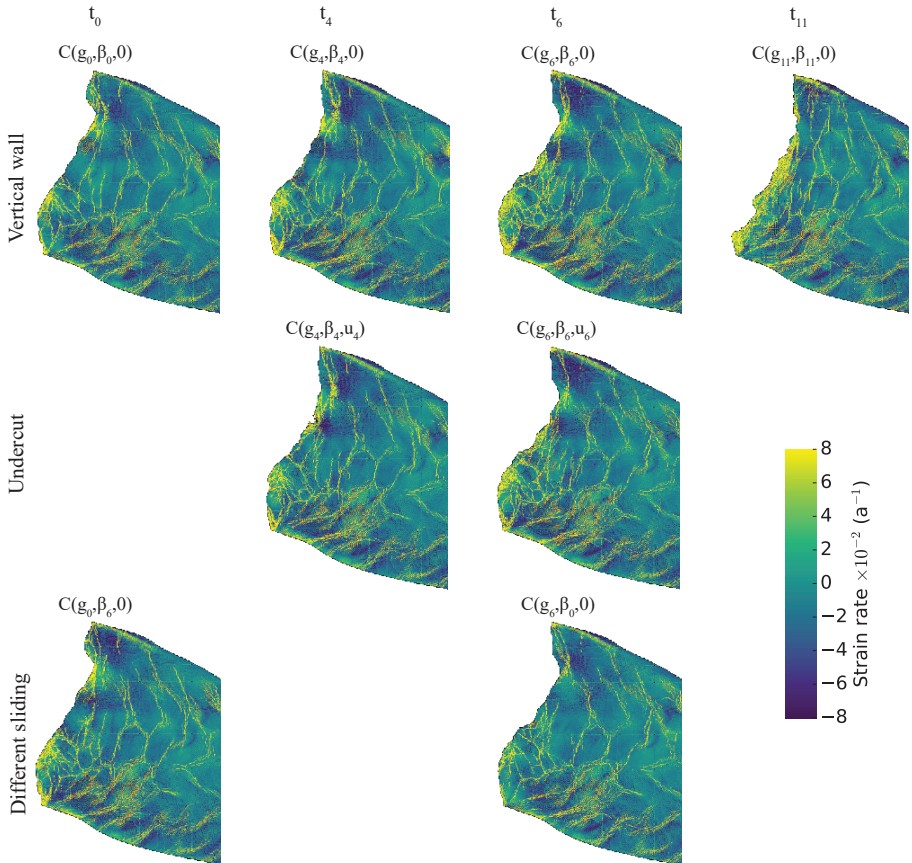

**Figure 11.** Strain rates modelled with HiDEM for each configuration. Yellow color shows the crevasse pattern and is denser close to the front where the difference between each configuration for the four selected iterations can be observed.

model based upon one-dimensional plume theory (e.g. Jenkins, 2011; Carroll et al., 2015; Slater et al., 2016) would be less computationally expensive and may allow some of these limitations to be addressed. However, such a model would not capture the strong surface currents driven by the plume which are important for the terminus morphology studied here.

For ND (Fig. 8b and d), the undercutting is in line with the observed front to a certain extent, particularly for $t_4$. However, for SD, apart for $t_3$, no apparent correlation between modelled undercutting and observed front location seems to exist. However, Fig. 10 shows that modelling calving with undercutting at SD and ND for $t_4$ and $t_6$ gives a good fit to observation. The difference in agreement with the observed front position and the modelled calving could possibly be explained by the uncertainty in discharge or the different character of the plume at high and low discharge. The low dependence of calving front position on modelled undercutting in situations of low discharge seems to have no major influence on the performance of the calving model. At Kronebreen, the high discharge relative to the shallow depth of the terminus drives strong gravity currents at the surface as water is rapidly exported horizontally away from the plume. The melt rates driven by these gravity currents are significant as shown in Fig. 7, and in some cases dominate over the melt rates driven by the plume at depth. The difference between low and high discharges is therefore slightly counter-intuitive. At low discharges, when maximum melt rates occur at depth, the terminus is more undercut but in a narrower area; meanwhile, at higher discharges, strong undercutting occurs but over a much wider area of the terminus. This suggests that calving behaviour may be very different in these two situations.

## 5.2 Calving model

Because the imposed undercuttings are the product of melt during the whole interval between observations, the model results should be treated with caution. Benn et al. (2017) compared HiDEM calving for specified undercuttings of different sizes and showed that calving magnitude increases with undercutting size. For small undercuttings, calving simply removes part of the overhang, but for large undercuttings calving removes all of the overhang plus additional ice. The mechanisms are different in each case: low-magnitude calving for small undercuttings occurs through collapse of part

of the unsupported overhang, whereas high-magnitude calving for large undercuttings involves forward rotation of the whole front around a pivot point located at the base of the undercut cliff. The long time-step intervals (11 or 18 days) between the starting geometry and the HiDEM simulation in the present study might therefore bias the results towards higher calving events. Testing this possibility is beyond the scope of the present paper, but remains an important goal for future research. Despite this caveat, our results compare well with observations, and yield valuable insights into the calving process.

Firstly, the HiDEM results show that undercutting associated with meltwater plumes is an essential factor for calving during the melt season ($t_4$ and $t_6$). Surface melt leads to the formation of a subglacial drainage system that ultimately releases the water into the ocean from discharge points at the front of the glacier. Simulations without frontal undercutting at these subglacial discharge locations do not agree well with observed frontal positions and mean volumetric calving rates. In contrast, simulations with frontal undercutting reproduce the retreat reasonably well at these locations, particularly where the discharge is high such as at ND. The largest discrepancy between modelled and observed calving is in the region south of SD at $t_4$. Here, the model predicts calving of a large block, whereas the observed front underwent little change. This largely reflects the rules used for calving in HiDEM: any block that is completely detached from the main ice body is considered as calved, even if only separated by a narrow crack from the rest of the glacier and still sitting at its original position. This is the case for the large 'calved' region south of SD at $t_4$, where the block may have been completely detached but remained grounded and in situ. If this were to occur in nature, it would not register as a calving event on satellite images. The discrepancy between model results and observations at this locality therefore may be more apparent than real.

Secondly, the model results replicate the observed high calving rates at $t_{11}$, after the end of the melt season when there is no undercutting. At this time, the observed mean volumetric calving rate is $24.99 \times 10^5 \, \mathrm{m}^3 \, \mathrm{d}^{-1}$, which compares well with the HiDEM rate of $28.50 \times 10^5 \, \mathrm{m}^3 \, \mathrm{d}^{-1}$. These values are much higher than those at the start of the melt season, when there is also zero undercutting. This contrast can be attributed to the high strain rates in the vicinity of the ice front at $t_{11}$, which would encourage opening of tensile fractures (Fig. 11). In turn, the high strain rates result from low basal friction (Fig. 5d), likely reflecting stored water at the glacier bed after the end of the melt season. It is possible that geometric factors also play a role in the high calving rates at $t_{11}$, because the mean ice front height is greater at that time than at $t_0$, reflecting sustained calving retreat during the summer months, which would have increased longitudinal stress gradients at the front (Benn et al., 2017). This interpretation is supported by experiments $C(g_0, \beta_6, 0)$ and $C(g_6, \beta_0, 0)$, in which the basal friction values are transposed for non-

undercut ice geometries at $t_0$ and $t_6$. Imposing low friction ($\beta_6$) at $t_0$ produces mean volumetric calving rates similar to (but smaller than) those observed at $t_6$, whereas imposing high basal friction ($\beta_0$) at $t_6$ produces low volumetric calving rates similar to those observed at $t_0$. The influence of basal friction on calving rates is consistent with the results of Luckman et al. (2015), who found that a strong correlation exists between frontal ablation rates and ice velocity at Kronebreen when velocity is high. Low basal friction is associated with both high near-terminus strain rates and high velocities, facilitating fracturing and high rates of ice delivery to the front. Our experiments do not include varying fjord water temperature, so we cannot corroborate the strong correlation between frontal ablation and fjord temperature observed by Luckman et al. (2015). However, our results are consistent with their finding that melt-undercutting is a primary control on calving rates, with an additional role played by ice dynamics at times of high velocity.

## 6 Conclusions

In this study, we use the abilities of different models to represent different glacier processes at Kronebreen, Svalbard with a focus on calving during the melt season of 2013. Observations of surface velocity, front position, topography, bathymetry and ocean properties were used to provide data for model inputs and validation.

The long-term fluid-like behaviour of ice is best represented using the continuum ice flow model Elmer/Ice that computes basal velocities by inverting observed surface velocities and evolves the geometry, including the front position. During the melt season, a subglacial hydrology system is created and the water is eventually evacuated at the front of the glacier. We used a simple hydrology model based on surface runoff directly transmitted to the bed and routing the basal water along the deepest gradient of the hydraulic potential. Two subglacial discharge locations have been identified by this approach: the northern one evacuates water with a high rate ($\sim 10$–$100 \, \mathrm{m}^3 \, \mathrm{s}^{-1}$) and the southern one with a low rate ($\sim 1$–$3 \, \mathrm{m}^3 \, \mathrm{s}^{-1}$). This fresh water is subsequently mixed with ocean water. Rising meltwater plumes entrain warm fjord water and melt the subaqueous ice creating undercuttings at the subglacial discharge location. We modelled the plume with a simplified 2D geometry using a high-resolution plume model based upon the fluid dynamics code Fluidity adapted to the front height and the ocean properties of Kronebreen. Melt rates depend on the discharge rate and the shape of the plume differs greatly with its magnitude. Higher discharges tend to let the plume rise to the surface close to which melt rates are the highest while low discharges concentrate the melt at lower elevations. The melt rates are then projected to the actual frontal geometry taking into account the subaqueous ice-front shape of the former timestep. It is interesting to note that modelled undercuttings

for high subglacial discharges are spatially close to the observed calving front whereas such a correspondence is not evident for small discharges. The elastic-brittle behaviour of the ice, such as crevasse formation and calving processes, is modelled using a discrete particle model, HiDEM. Two factors impacting glacier calving are studied here using HiDEM: i) melt-undercutting associated with buoyant plumes; and ii) basal friction, which influences strain rates and velocity near the terminus. The performance of the calving model is evaluated quantitatively by comparing observed and modelled mean volumetric: calving rate and qualitatively by comparing calved regions. Results show that modelled calving rates are smaller than observed values during the melt season in the absence of melt-undercutting, and that there is a closer match with observations if undercutting is included. Additionally, there is good agreement between modelled and observed calving before ($t_0$) and after ($t_{11}$) the melt season, when there is no undercutting. Both modelled and observed calving rates are much greater after the melt season than before, which we attribute to lower basal friction and higher strain rates in the near-terminus region at $t_{11}$. The influence of basal friction on calving rates is corroborated by model experiments that transposed early and late-season friction values, which had a large effect on modelled calving. These results are consistent with the conclusions of Luckman et al. (2015), that melt-undercutting is the primary control on calving at Kronebreen at the seasonal scale, whereas dynamic factors are important at times of high velocity (i.e. low basal friction).

In this paper, we have shown that offline coupling of ice-flow, surface melt, basal drainage, plume-melting, and ice-fracture models can provide a good match to observations and yield improved understanding of the controls on calving processes. Full model coupling, including forward modelling of ice flow using a physical sliding law, would allow the scope of this work to be extended farther, including prediction of glacier response to atmospheric and oceanic forcing.

*Author contributions.* DV contributed to the design of the study, the offline coupling, the development of the undercutting model, the Elmer/Ice and HiDEM set-ups and the writing of the manuscript. DB edited the manuscript. All other authors provided comments to the manuscript. JÅ developed the HiDEM model and used Kronebreen as test and development case. AE developed the plume model. TZ contributed to the Elmer/Ice set-up. RP calculated the water discharge. DB and AL provided the observed surface velocity maps. WVP developed the coupled energy balance - snow modelling approach. JK provided the interpolated bed map.

*Acknowledgements.* We thank CSC - IT Center for Science Ltd. for the CPU time provided under Nordforsk NCoE SVALI. Thomas Zwinger was supported by the Nordic Center of Excellence eSTICC (eScience Tools for Investigating Climate Change in Northern High Latitudes) funded by Nordforsk (grant 57001). Acquisition of the TerraSAR-X imagery was funded by the ConocoPhillips Northern Area Program, via the CRIOS project (Calving Rates and Impact on Sea Level). The leading author received an Arctic Field Grant from the Svalbard Science Forum to acquire radar lines for the basal topography in 2014. Finally we would like to thank the reviewers and the editor for their input and help to improve the paper.

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
