# Peer review of "Effects of undercutting and sliding on calving: a global approach applied to Kronebreen, Svalbard"

_The Cryosphere, 2017_

## Referee Comment (RC1) · Anonymous Referee #1 · 21 Sep 2017

**The Cryosphere TC2017-166** *"Effects of undercutting and sliding on calving: a coupled approach applied to Kronebreen, Svalbard"* by Vallot and others.

A suit of 6 models is used all together to study the effect of undercutting and sliding on calving using the dataset of Kronebreen in Svalbard: (1) the surface mass balance and surface runoff are obtained from a coupled energy balance - snow model ; (2) water is routed through the glacier front using a basal hydrology model assuming zero effective pressure ; (3) at the front, melt distribution is inferred by plume model using the code Fluidity ; (4) undercutting is accounted by transforming the front surface,

initially vertical, according to the plume melt distribution ; (5) ice velocity, isotropic pressure and the glacier geometry are determined from the Stokes equation using the Elmer/Ice code ; and (6) calving is determined using a discrete particules model (HiDEM). All the models are not fully coupled and more often used in a one-way coupling.

One of the main conclusion of the paper is that calving rate is controlled by basal sliding. I can see two problems in the methodology that question the validity of this conclusion. First, the friction coefficient inferred using inverse methods with Elmer/Ice has to be scaled down by "some" orders of magnitude when used with HiDEM. I didn't understand the justification regarding the different time scales of calving and sliding processes to justify this scaling. The "some orders of magnitude" should be quantified. Is it a constant for the whole domain? Is it the same value (give it) for all simulations? This should be explained much more precisely. Second, calving rate is a continuous view of calving, as calving is discrete and a calving rate can only be inferred when averaging a number of calving events during a given time. Here, it seems that calving rate is inferred from one simulation of HiDEM and I then suppose that it is inferred from one calving event? Or a limited number of calving events arising during a very short time? Can we deduce a calving rate from that, and then conclude that calving rate is very sensitive to basal sliding? Also, it should be worse verifying that the result are not too strongly dependent on the time step in between two HiDEM simulations. How different are the calving rates obtained by running the HiDEM model every $dt$, $dt/2$, $dt/4$ time step of the Elmer/Ice model?

**Minor remarks:**

page 2, lines 12-15: The fact that it is untested against observations certainly also apply to the particule models (or you should give a reference in which the

particules model is validated against data). The distinction between continuous and discrete approaches could be a bit more rigorous and objective. There are also some drawbacks in the particules model that will anyway render its use very difficult for large or long applications.

page 2, line 22: with the discrete element model HiDEM

Figure 1: what are the different colours? Especially the white versus grey?

page 3, line 10: as shown by Nahrgang et al. (2014) presenting (there are similar problems with the use of brackets for the references all along the manuscript. Please, check this).

Figure 2: should be Elmer/Ice not Elmer/ICE to be consistent with the text.

Table 1: give in the first column the number of day $t_0 = 0$, $t_1 = 11$ d, etc...

page 6, lines 1-2: here you are mentioning the one-way coupling between HiDEM and Elmer/Ice, and then saying that a completely coupled model would also couple the hydrology and the ice flow. But to be completely coupled, you should add the coupling with the plume model? I would suggest to modify the transition: *Also, an improvement could be to calculate the friction...*

page 6, line 20: in this part it should be clearly mentioned what is making the front position advance or retreat. Which equation is solved for the front position? Is it a similar to equation (2) and therefore the front is moving as a balance of ice flow and front melting?

page 7, line 15: on which Grid? The finite element one? Why not solving equation (3) using the finite element method?

page 7, line 19: I cannot understand what you mean by this last sentence... what is flow accumulation?

page 8, line 10: that sea level corresponds to $z = 0$ is already mentioned above.

page 9, line 5 and after: this part is not clear. What are the reasons for these 3 different treatments should be explained.

page 9, line 11: again, same remark as above: which method? Are you solving a free surface evolution equation for the front? Is the Elmer/Ice model time step also 11 days? This should be specified somewhere.

page 9, line 27: that varies spatially according to the inversion done using Elmer/Ice?

page 9, line 32: see my main point. It clearly questions the fact that a calving rate can be inferred from this approach? What does it change if you run HiDEM every two (or half) timestep?

page 10, line 4: which complication? As for the "some orders of magnitude", the explanation should be more precise.

Table 2, last line fisrt column : $C($ instead of $C_($

Figure 5: Downstream the front, one would expect zero friction? Are the value on this plot extrapolated? Should be mentioned.

Figure 6: I would expect that the discharge increase along the water path and I don't see this from the plot. On (b), the axis for SD should also start a 0 (and with a continuous curve going to 0 for no discharge).

Table 3: how the sum of SD and ND volume compare to the integrated runoff over the basin?

Figure 8; what represent the horizontal thin lines in the ocean?

[Figure]

page 15, line 6: To my understanding, $a_c^{obs}$ doesn't include only calving but also melt at the front? So, it should also be mentioned.

Table 4: it should be $a_m$ and not $a_m^{obs}$ in the table? In the legend, I am a bit confused by what you call the tangential ice velocity (tangent to the front?). Isn't it the velocity normal to the front that you mean here? Same in the legend of Fig. 9.

Figure 11: As you mention in the text that Fig. 11 shows strain rates that ressemble crevasses pattern, would be nice to have an aerial image of the real crevasse pattern? How do you explain the very similar patterns for all simulations inside the domain? What drive these features? And why choosing to plot strain-rates when you could directly plot places where bounds are broken?

page 20, line 1: regarding the key role of basal friction, see my main comment

page 22, line 7: Elmer/Ice

---

## Referee Comment (RC2) · Anonymous Referee #2 · 4 Oct 2017

**1   Summary statement**

The manuscript by D. Vallot et al. studies the ice front evolution during the melt season of Kronebreen glacier, Svalbard. The authors use a global approach that combines different models computing ice flow evolution, surface mass balance, subglacial hydrology, plume model, undercutting model and fracture dynamics, to assess the calving of this glacier. They study its seasonal evolution and compare modeled results with observations. They also investigate the impact of several parameters on the location and rate of calving, and conclude that the glacier geometry controls the location of calving, that the basal sliding controls its rate, and that undercutting is necessary to reproduce

observations.

This manuscript presents an interesting study that not only focuses on one aspect of calving (e.g., fracture, undercutting) but also combines many different processes that are known to affect calving. Modeled results are compared with observations, and experiments are performed to assess the influence of several parameters. However, the manuscript is not always clear, and seems to have been put together in a rush. There are inconsistencies between some variable names (e.g., $C\left(t_i, \beta_i, u_i\right)$ and $C\left(g_i, \beta_i, u_i\right)$), and symbols that are used for two different variables (e.g. $u$ for both velocity and undercut). It therefore needs some major clean up to be clarified and ensure consistency in notations (see detailed remarks below). Furthermore, some choices made are not explained or justified. This is the case for the reduction in basal friction for the discrete particule model, or the decision to keep or remove the undercut depending on the ice front location relative to the previous position and current observation. These decisions are not physically justified, and the impacts of such choices are not studied. Finally, some conclusions are also not well supported by the analysis. One important result is that "the geometry controls the calved zones while the basal friction (glacier dynamics) controls the magnitude of calving (calving rate)". However, the impact of the basal friction on Fig. 10 (a and c only show the impact of friction) is not clear, as the calving rate is not very different for the high and low friction scenarios, and adding undercutting (Fig. 10c) has an impact similar to changing basal friction. So if these three parameters (geometry, basal friction, and undercutting) have an impact on calving, more experiments are needed to conclude on the specific impact of each individual parameter.

**2 Major comments**

The title of this paper and subsequent references to "coupling" are misleading. There is no coupling performed in this model. The different models used to represent the calving front processes are put together and outputs from one model are inputs of other models, but there is no coupling. Fig. 2 illustrates this very well: arrows all have the same direction and outputs from HiDEM are never used as inputs for other models. What this paper does is provide a comprehensive approach to the question of calving, and I think a title using "global approach" or something similar would be more accurate.

As mentioned above, the conclusions separating the impact of geometry, basal sliding and undercutting are not well supported by the results provided. Looking at Fig. 10, it seems that all parameters have an impact on both the location and extend of retreat, but they cannot be clearly distinguished without further experiments.

It is not clear if all the 11 time steps described in Tab. 1 are modeled, or if only a subset of these times are used. Results from $t_0$, $t_4$, $t_6$ and $t_{11}$ are mostly presented, but Fig. 10 also shows results at different time steps.

What is the rational for keeping or removing the undercut in one case or another when the ice front advances or retreats (Fig. 4 and p.9)? Some explanations justifying these choices should be added as opposed to presenting the choices made without any justification. I cannot quite figure out why the undercut from the previous profile is not always considered.

**3 Line by line comments**

p.1 l.17: "rigorous methods": the problem is not so much about rigorous methods but more about some processes impacting calving that we still don't understand, as well as

small scale features (mm long cracks) that cannot be observed and included in models.

p.1 l.20: "impacting on submarine melt rate" → 'impacting submarine melt rate"

p.2 l.1: "during the summer and the autumn" → "during summer and autumn"

p.2 l.3: "followed by ice-front collapse": not clear

p.2 l.12: The problem is actually not so much the representation of calving in models but the processes impacting calving that are not enough understood and therefore cannot be included into models.

p.2 l.16: Again here, it is not really coupling but feeding the particule model with appropriate inputs from Elmer/Ice.

p.2 l.32: "one of the fastest" → "one of the fastest glacier"

p.2 l.33: How much seasonal variation is this glacier experiencing?

p.3 l.3: How large are the seasonal variations? What is the velocity in winter?

p.3 Fig.1: Consider adding Kongsfjorden on the figure. Calving front position for 16 Octobre 2013 is not visible, consider changing it.

p.3 l.10 "by (Nahrgang et al., 2014)" → "by Nahrgang et al. (2014)"

p.3 l.11-15: Past tense should be used to describe measurements made in 2013.

p.3 l.11-15: How representative of the seasonal cycle are these values?

p.4 Fig.2: There is no feedback and therefore no coupling shown on this figure. Outputs from one model are used as parameters/inputs for the next model.

p.4 l.7: What about the other observations (geometry, ice temperature and viscosity)? Where do they come from?

p.5 l.6: Is the sliding inverted just at the beginning of the simulation or recomputed for each time step? In this case how is the change in the glacier geometry computed (or

maybe observations are used)?

p.5 l.9: How are the front and surface evolved? Are they run from the previous iteration and therefore the 12 time steps are run with the model? In this case, why only show results for 6 cases and not the entire melt season? If not, how are the front and surface evolved? Also, modeling ice front changes in ice flow model is not an easy task and is currently the subject of active research. How is the front evolved with the Elmer/Ice model? There is no reference or explanation of how the ice front migrates and no Elmer/Ice paper describing such an evolution to my knowledge. This has to be better explained.

p.6 l.1: "coupling": same as above

p.6 l.1: If the front position is not used as inputs for the Elmer/Ice initial front position, what is used then?

p.6 Eq.1: Consider using vectors. Also $u$ is used both here for the velocity, and later (e.g. Tab.2) for the undercutting. Change one or the other.

p.6 l.8: Again here, is the friction optimized at each time step or just at the beginning of the simulation?

p.6 l.9: "the self-adjointness" → "a self-adjoint algorithm"

p.6 l.10 and l.11: Consider adding older references that first used such methods.

p.6 l.13: This paragraph could be put in the data section (section 3.1) to improve consistency.

p.6 l.20: "The front position is also able to advance": How is it able to advance? See point above

p.6 l.20: "$F_i^{obs}(0)$": I would imagine that observations show the front position on the surface of the glacier and not at sea level.

p.6 l.20-21: There are several front positions observed and computed. The authors should start by listing all the front position computed ($F^{elmer}$, $F^{HiDEM}$, ...) and explaining where they come from. That might be something to add on Fig.2.

p.6 section 3.3: What is the resolution (horizontal and vertical) of the model, especially close to the ice front? What are the time steps used for the continuum model?

p.7 l.8: convention for the reference (twice)

p.7 l.29: "five kilometers" → "five kilometers away"

p.8 l.1: How long does it take to reach a steady-state?

p.8 l.4-7: So my understanding is that the discharge varies but not the ocean conditions. Ocean conditions are reported quite accurately on p.3, so why not use these conditions instead of uniform ambient ocean properties? Also, in all these cases, the ice front is assumed to be vertical, why not try cases with pre-existing undercutting? I understand that it might not be possible to test all these cases, but at least assessing the uncertainty caused by such assumptions would be important.

p.9 l.6-10: What is the rational for keeping or removing the undercut in one case or another? Some explanations justifying these choices should be added.

p.9 l.19: How many broken beams are added and how was this number chosen? What is the impact of increasing or reducing this number on the results? Also does the number of broken beams increase during the melt season as the ice gets more damaged?

p.9 l.30: How long is the HiDEM model run for at each time step? And how long does it take to run it?

p.10 l.2: What kind of instabilities are developing and why?

p.10 l.7-11: What is the rational for decreasing the friction? How is the choice of friction impacting your results?

p.10 l.19: It should be mentioned that this is volumetric ablation rate (same for volumetric calving rate in the rest of the paper). Many people use calving/ablation rate as changes per unit area (in m/yr), which can be confusing.

p.10 l.20: Integrals over $Gamma$ usually refer to contour intervals and not surface integrals, using $S$ or $\Sigma$ instead would be more consistent with literature.

p.10 Eq.5: What is $z\Gamma_w$?

p.10 l.28: "parameterisations" → "parameters"

p.10 l.30: $u$ was already used for velocity (see above)

p.10 l.30: Only 4 time steps are mentioned here. What happens to the other ones, are they just excluded? In this case, what is used for the prior undercut?

p.11 l.1: Only a subset of $(i, j) \in [0, 4, 6, 11]$ is covered, not accurate.

p.11 l.6: configuration $C_k$ is not defined and not used anywhere else, should be consistent with the rest of the paper

p.11 Tab.2: Configuration is here a function of time ($t_i$) as opposed to geometry ($g_i$) in the rest of the paper

p.12 Fig.6 caption: "data gaps corresponds" → 'data gaps correspond"

p.13 l.Tab.3: Discharged should be provided in $m^3/s$ to be consistent with the rest of the text. No data between $t_{10}$ and $t_{11}$, this should be added even if the values are just zero. Also, how are the melting rates for each case computed based on Fig.7? Is an interpolation between the four cases been performed? Or something else?

p.13 Fig.7: How different are the results if there is undercut introduced in the geometry?

P.14 Fig.8: It is the only time in the paper, where results from times other than $t_0$, $t_4$, $t_6$ and $t_{11}$ are presented. Are the other time steps computed? And what is the rational to only present some ice front positions here?

p.14 Fig.8: If $z$ is the height above sea level, Fig.8 b and c are for $z = -3m$ and $z = -42m$, and the stars in Fig.8 d and e indicate the plan view elevation, why are the start not aligned at the same height on Fig.8 d and e? Also it might be more clear to use "Elevation from sea level" or something similar instead of "Distance to the bed" on Fig.8 d and e as this is what is used in the rest of the figure.

p.14 l.11: What do the authors mean by "smoother"?

p.14 l.12: "stretching" → "stretches"

p.15 l.2: Why not do it more often? How long does it take to run the model?

p.15 Tab.4: "modelled" → "estimated": the melt is estimated from observations, not modeled. What is $\dot{a}_m^{obs}$? Also add zero values where appropriate instead of leaving empty spaces.

p.16 Fig.9: $\dot{a}_c = \dot{a}_{c,u} - \dot{a}_{c,L}$ on p.15 (minus not plus), so it's not clear what is shown on this figure.

p.19 l.5: "Due to this, there is some uncertainty" → "There is therefore some uncertainty"

p.19 l.7: Add references (or something else) to justify this statement that is not supported.

p.19 l.24: Is that what is observed by the authors?

p.20 l.1: "no retreat at all": this is not supported by the model results, there is some retreat in the southern part of the domain.

p.20 l.9-10: I don't think that the experiments made and results support such a conclusion, the role of sliding and geometry cannot be clearly separated.

p.20 l.19: "the velocity higher" → "the higher velocity"

p.20 l.19: "seem" → "seems"

p.20 l.30: "reproduces" → "reproduced"

p.21 l.12: "in 2D" → "with a simplified 2D geometry"

p.21 l.14: How do the melt rates compare to previous results?

p.21 l.23: What is referred to as the calving model?

p.22 l.5: "would be implemented" → "were implemented"

---

## Author Comment (AC1) · 3 Nov 2017

We first want to thank the referee for the constructive comments. Our answered answers to the questions are detailed below.

**General comment:**

*One of the main conclusions of the paper is that calving rate is controlled by basal sliding. I can see two problems in the methodology that question the validity of this conclusion. First, the friction coefficient inferred using inverse methods with Elmer/Ice has to be scaled down by "some" orders of magnitude when*

*used with HiDEM. I didn't understand the justification regarding the different time scales of calving and sliding processes to justify this scaling. The "some orders of magnitude" should be quantified. Is it a constant for the whole domain? Is it the same value (give it) for all simulations? This should be explained much more precisely.*

This is a good point and our original text was not well formulated to explain this properly: There is a clear separation of timescales between the velocities of sliding ($\sim$ m/day) and calving ice ($\sim$ m/sec). This means that as long as sliding is slow enough to be negligible during a single calving event, we can change it without much effect on any single calving event. As an approximation we can assume that fast processes are at equilibrium when we consider slower timescales. However, a rescaling speeds up the frequency of calving, and we can thus 'speed up', within reason, the few minutes of HiDEM simulation to effectively model calving which would otherwise take tens of hours or days, and thus be practically impossible to simulate with HiDEM. By applying scaling, the calving events modelled during the simulation of HiDEM (few minutes) correspond to the sum of calving events that would happen during the time scale of sliding. The scaling factor that we use is the same for the whole domain and for all simulations. We use a friction scaling factor for $\beta$ equal to $10^{-2}$ (or sliding velocity scaled up by $10^2$), and simulations run until calving stops and a new quasi-static equilibrium is reached.

This is now explained in the text (p. 12, lines 10–18): "This means that as long as sliding is slow enough to be negligible during a single calving event, we can change it without much effect on any single calving event. As an approximation we can assume that fast processes are at equilibrium when we consider slower timescales. However, a rescaling speeds up the frequency of calving, and we can thus 'speed up', within reason, the few minutes of HiDEM simulation to effectively model calving which would otherwise take tens of hours or days, and thus be practically impossible to simulate with HiDEM. By applying scaling, the calving events modelled during the simulation of HiDEM (few minutes) correspond to the sum of calving events that would happen

during the time scale of sliding. The scaling factor that we use is the same for the whole domain and for all simulations. We use a friction scaling factor for $\beta$ equal to $10^{-2}$ (or sliding velocity scaled up by $10^2$), and simulations run until calving stops and a new quasi-static equilibrium is reached."

*Second, calving rate is a continuous view of calving, as calving is discrete and a calving rate can only be inferred when averaging a number of calving events during a given time. Here, it seems that calving rate is inferred from one simulation of HiDEM and I then suppose that it is inferred from one calving event? Or a limited number of calving events arising during a very short time? Can we deduce a calving rate from that, and then conclude that calving rate is very sensitive to basal sliding?*

This is another good point: During one simulation with HiDEM, several calving events are triggered. Calving rate is then inferred from the difference between the initial and final position of the front, after calving has stopped and the glacier has come to rest. This approach is, as the referee points out, dependent on several assumptions, and the error is not easy to estimate. Therefore, the comparison with observed calving rate is very important in order to estimate the validity of this approach. To be clearer, we use "mean volumetric calving rate" instead of just "calving rate". The discussion on basal sliding is based on the comparison between two time steps with the assumptions made in the simulations. Of course, the fact that it is a one-way coupling and that the advance is step-wise and not continuous (due to observation time resolution and modelling limitations) makes the conclusion dependent on these assumptions. We have now changed the text to better explain that the model results regarding basal sliding and calving are valid only under the specific model assumptions.

In Methods/Frontal ablation calculation:

"For the modelled case, during one simulation with HiDEM, several calving events are triggered. Volumetric calving rate is then inferred from the difference between the initial,

$F_i^{elmer}$, and final position, $F_i^{hidem}$, of the front, after calving has stopped."

***Also, it should be worth verifying that the results are not too strongly dependent on the time step in between two HiDEM simulations. How different are the calving rates obtained by running the HiDEM model every dt, dt/2, dt/4 time step of the Elmer/Ice model?***

This is yet another good point that would need further investigation in a fully coupled version. Here we only have observations every 11 days and do not know what happens at dt/2 or dt/4 so it would not be possible to compare. However, we did early test with gradually vanishing sliding and vanishing under-cut. When sliding vanishes, the under-cut determines calving (compare with Benn et al., 2017), and when both vanish, there is typically no calving at all. When comparing HiDEM calving for specified undercuts of different sizes in Benn et al (2017), the results shows that the magnitude of calving increases with undercut size: for small undercuts calving just removes part of the overhang, but for large undercuts calving removes all of the overhang plus additional ice. The mechanisms are different in each case - low-magnitude calving for small undercuts just involves collapse of part of the unsupported overhang, whereas high-magnitude calving for large undercuts involves forward rotation of the whole front around a pivot point located at the base of the undercut cliff. When the time step is changed as the referee suggest above, there are differences, but the overall rate changes stays roughly within $\pm 50\%$.

We have now changed the following text to better explain this. In Methods/ Calving with first-principles ice fracture model HiDEM

"If the time step is changed, the overall rate change stays roughly within $\pm 50\%$"

In the discussion:

"Because the imposed undercuts are the product of melt during the whole interval between observations, the model results should be treated with caution. Benn et al.

(2017) compared HiDEM calving for specified undercuts of different sizes and showed that calving magnitude increases with undercut size. For small undercuts, calving simply removes part of the overhang, but for large undercuts calving removes all of the overhang plus additional ice. The mechanisms are different in each case: low-magnitude calving for small undercuts occurs through collapse of part of the unsupported overhang, whereas high-magnitude calving for large undercuts involves forward rotation of the whole front around a pivot point located at the base of the undercut cliff. The long time-step intervals (11 or 18 days) between the starting geometry and the HiDEM simulation in the present study might therefore bias the results towards higher calving events. Testing this possibility is beyond the scope of the present paper, but remains an important goal for future research. Despite this caveat, our results compare remarkably well with observations, and yield valuable insights into the calving process."

**Minor remarks:**

***page 2, lines 12-15: The fact that it is untested against observations certainly also apply to the particle models (or you should give a reference in which the particle model is validated against data).***

This is true but this is also the subject of the paper: to test HiDEM results against observations. However, the sentence "These are largely untested against observations, and may fail to adequately represent key processes." was not well placed and broke the transition between the former and the next sentence. Therefore we removed it.

***page 2, lines 12-15: The distinction between continuous and discrete approaches could be a bit more rigorous and objective. There are also some drawbacks in the particle model that will anyway render its use very difficult for large or long applications.***

We gave more details on the discrete models and drawbacks to transition to the next paragraph:

"These problems can be circumvented using discrete particle models, which represent ice as assemblages of particles linked by breakable elastic bonds. Ice is considered as a granular material and each particle obeys Newton's equations of motion. Above a certain stress threshold, the bond is broken, which allows the ice to fracture. Åström et al. (2013, 2014) showed that complex crevasse patterns and calving processes observed in nature can be modelled using a particle model, the Helsinki Discrete Element Model (HiDEM). Bassis and Jacobs (2013) used a similar particle model and suggested that glacier geometry provides the first-order control on calving regime. However, the drawback of these models is that due to their high computer resource demand, they only can be applied to a few minutes of physical time. A compromise should be found by coupling a continuum model, such as Elmer/Ice, to a discrete model, such as HiDEM, to successively describe the ice as a fluid and as a brittle solid."

*page 2, line 22: with the discrete element model HiDEM*

Done!

*Figure 1: what are the different colours? Especially the white versus grey?*

We added more details in the caption:

"Ocean is in blue, bare rock is in red and glacier ice is in white. The grey area represents the Kronebreen glacier system."

*page 3, line 10: as shown by Nahrgang et al. (2014) presenting (there are similar problems with the use of brackets for the references all along the manuscript. Please, check this).*

We changed it there and at other places.

*Figure 2: should be Elmer/Ice not Elmer/ICE to be consistent with the text.*

We changed it.

*Table 1: give in the first column the number of day = 11d, etc...*

Done!

*page 6, lines 1-2: here you are mentioning the one-way coupling between HiDEM and Elmer/Ice, and then saying that a completely coupled model would also couple the hydrology and the ice flow. But to be completely coupled, you should add the coupling with the plume model? I would suggest to modify the transition: Also, an improvement could be to calculate the friction...*

We add more information as suggested:

"We call this approach a one-way coupling because inputs to the HiDEM are output results from Elmer/Ice and undercutting model but not vice versa. In Elmer/Ice, we use the observed frontal positions. A completely coupled physical model would use the output of the HiDEM, the modelled front position, as input to the ice flow model Elmer/Ice and the undercutting model. It would also calculate the basal friction from a sliding law rather than an inversion. In principle, such an implementation is possible using the same model components as this study."

*page 6, line 20: in this part it should be clearly mentioned what is making the front position advance or retreat. Which equation is solved for the front position? Is it a similar to equation (2) and therefore the front is moving as a balance of ice flow and front melting?*

The front is advanced by imposing a Lagrangian scheme over a distance equal to the ice velocity multiplied by the time step. We do not account for the melting during the advance because we only have observations at the beginning and the end of each timespan. Instead, we lump frontal melting by applying an undercut after the advance as explained hereafter.

We add this information.

"The front is advanced by imposing a Lagrangian scheme over a distance equal to the ice velocity multiplied by the time step. We do not account for the melting during the

advance because we only have observations at the beginning and the end of each timespan. Instead, we lump frontal melting by applying an undercut after the advance as explained hereafter."

***page 7, line 15: on which Grid? The finite element one? Why not solving equation (3) using the finite element method?***

It is possible to solve equation (3) using the finite element method but not the D-infinity flow method. Of course, we could have used another subglacial hydrology model but this was not the case for this study. It is the surface runoff grid ($100\times100$ m).

We add this information in the description of the surface runoff and of the flow path calculation:

"Surface runoff is modelled on a $100\times100$ grid."

"We use the surface runoff grid."

***page 7, line 19: I cannot understand what you mean by this last sentence... what is flow accumulation?***

A matlab routine is calculating the flow direction from Equation (3) and given D-infinity flow method. The accumulated flow is the sum of all water flowing in a cell from adjacent cells given this flow direction. Each cell is weighted by the surface runoff calculated for the cell.

***page 8, line 10: that sea level corresponds to z = 0 is already mentioned above.***

We remove the sentence.

***page 9, line 5 and after: this part is not clear. What are the reasons for these 3 different treatments should be explained.***

The three different treatments depend on the relative position between the observed and modelled front. To explain it better, we added more details before explaining the

cases:

"When the first discharge occurs, the melt rate calculated with the plume model in 2D is summed for the period of time between $t_0$ and $t_1$ and projected to the advanced front $F_1^{elmer}(z = 0)$ (advanced from $F_0^{obs}(z = 0)$) at the location of the subglacial outlets and ice is removed normal to the front. This yields a new position of the front at depth $z$ below sea level called $F_1^{elmer}(z)$. At the second iteration, $t_2$, we know where the front would be if there had not been any calving between $t_1$ and $t_2$: $F_2^{elmer}(z = 0)$, which is the advanced front from the observed position at $t_1$, $F_1^{obs}(z = 0)$. So we can transfer the whole undercut from previous iteration to $F_2^{elmer}(z)$ if $F_1^{obs}(z = 0)$ is situated in front of $F_1^{elmer}(z)$ (see Fig. 4b–c). Otherwise, the undercut would have been fully or partly calved away (see Fig. 4b–c). We then apply the new undercut on this new geometry given the melt rates between $t_1$ and $t_2$.

At time $t_i$, the modelled front position at depth $z$ (advanced by Elmer/Ice from the observed front position at $t_{i-1}$) is $F_i^{elmer}(z)$ and the observed front position is $F_i^{obs}(z = 0)$. We advance this observed front with Elmer/Ice during $\Delta t = t_{i+1} - t_i$ to obtain the front position $F_{i+1}^{elmer}(z = 0)$ at $t_{i+1}$. We want to determine $F_{i+1}^{elmer}(z)$ and depth $z$ given the melt rate calculated between $t_i$ and $t_{i+1}$ and the state of the undercut from the previous front $F_i^{elmer}(z)$ updated by the observed front $F_i^{obs}(z = 0)$."

***page 9, line 11: again, same remark as above: which method? Are you solving a free surface evolution equation for the front? Is the Elmer/Ice model time step also 11 days? This should be specified somewhere.***

We do not use Elmer/Ice for the undercutting model and we do not use a free surface evolution equation. We use the front position evolved with Elmer/Ice, $F_i^{elmer}(z = 0)$ and apply the undercut on it. We first determine the melt rate during the time period between $t_i$ and $t_{i+1}$ given the accumulated discharge calculated by the hydrology model and using the plume model. Second, we project this melt rate onto the calving front using the method described above.

*page 9, line 27: that varies spatially according to the inversion done using Elmer/Ice?* Yes, we use the coefficient of friction calculated by the inversion with Elmer/Ice but scaled down as explained after.

*page 9, line 32: see my main point. It clearly questions the fact that a calving rate can be inferred from this approach? What does it change if you run HiDEM every two (or half) timestep?*

Please, refer to the general comment answer.

*page 10, line 4: which complication? As for the "some orders of magnitude", the explanation should be more precise.*

The complications induced come from the construction of the ice in HiDEM. It reads an input file with surface, bed and basal ice coordinates. The basal ice different from the bed where there is undercutting. It is not possible to add more ice under this basal ice and create an ice foot. We give more details:

"HiDEM reads a file with surface and bed coordinates on a grid and a file with surface and basal ice (to take into account the undercut) coordinates. When simulating with an undercut at a discharge location and in order to avoid complication in the HiDEM (position of the basal ice), we remove particles below the maximum melt (no ice foot)."

*Table 2, last line first column :* $C($ *instead of* $C_($

Done!

*Figure 5: Downstream the front, one would expect zero friction? Are the value on this plot extrapolated? Should be mentioned.*

We added more details in the HiDEM description: "In the ocean, the basal friction coefficient is extrapolated downstream of the front and taken equal to the mean of the values further from the terminus in case the ice advances."

*Figure 6: I would expect that the discharge increase along the water path and I*

*don't see this from the plot. On (b), the axis for SD should also start a 0 (and with a continuous curve going to 0 for no discharge).*

It does increase but it is maybe not so obvious with the logarithmic color scale. We use this colorscale to highlight the path. We changed the figure according to the comment (Fig. 1 from the answer).

*Table 3: how the sum of SD and ND volume compare to the integrated runoff over the basin?*

The sum of SD and ND should be actually equal to the integrated runoff over the basin.

*Figure 8; what represent the horizontal thin lines in the ocean?*

Do you mean in (d) and (e)? It represents the sea level. We add this information in the figure caption. "Horizontal lines in (d, e) represent the sea level for each time step."

*page 15, line 6: To my understanding, $a_c^{obs}$ doesn't include only calving but also melt at the front? So, it should also be mentioned.*

The melt at the front is not included in the calculation of $a_c^{obs}$ (see Eq. 4) but shown in am because it is modelled.

*Table 4: it should be am and not $a_m^{o}bs$ in the table? In the legend, I am a bit confused by what you call the tangential ice velocity (tangent to the front?). Isn't it the velocity normal to the front that you mean here? Same in the legend of Fig.9.*

This is true, we change to $a_m$.

We understand the confusion. It is actually normal to the front and tangent to the flow. It depends in which domain we are. Better clarification is given: "[. . .] difference between the tangential (ice flow direction) ice velocity at the front [. . .]"

*Figure 11: As you mention in the text that Fig. 11 shows strain rates that ressem-*

*ble crevasses pattern, would be nice to have an aerial image of the real crevasse pattern? How do you explain the very similar patterns for all simulations inside the domain? What drive these features? And why choosing to plot strain-rates when you could directly plot places where bounds are broken?*

We add a crevasse map in Figure 1 (Fig. 2 from the answer. The crevasses form as a result of increasing strain-rate towards the terminus, and as a result of shear strain near margins, which is rather similar in all cases. Strain-rate is better than broken bonds, since strain-rate differentiates between very narrow cracks and wide crevasses.

*page 20, line 1: regarding the key role of basal friction, see my main comment*

Please, refer to the general comment answer. We also change the whole section 5.2 to:

[revised manuscript text omitted]

*page 22, line 7: Elmer/Ice.*

Done!

[Figure]

[Figure]

**Fig. 1.**

[Figure]

0      1 km

**Fig. 2.**

---

## Author Comment (AC2) · 3 Nov 2017

We first want to thank the referee for the constructive comments. Our answers to the questions are as follows.

**Major comment:**

*The title of this paper and subsequent references to "coupling" are misleading. There is no coupling performed in this model. The different models used to represent the calving front processes are put together and outputs from one model are inputs of other models, but there is no coupling. Fig. 2 illustrates this very*

*well: arrows all have the same direction and outputs from HiDEM are never used as inputs for other models. What this paper does is provide a comprehensive approach to the question of calving, and I think a title using "global approach" or something similar would be more accurate.*

This is true and that is why we referred to it as a one-way coupling throughout the text. However, and the referee is right, the title is misleading and we change it to the suggestion: "Effects of undercutting and sliding on calving: a global approach applied to Kronebreen, Svalbard"

As mentioned above, the conclusions separating the impact of geometry, basal sliding and undercutting are not well supported by the results provided. Looking at Fig. 10, it seems that all parameters have an impact on both the location and extend of retreat, but they cannot be clearly distinguished without further experiments. It is true that more experiments are always better and we understand that the conclusion should acknowledge the fact that they are only valid for these cases. However, we do not agree with the statement in the summary "the impact of the basal friction on Fig. 10 (a and c only show the impact of friction) is not clear, as the calving rate is not very different for the high and low friction scenarios, and adding undercutting (Fig. 10c) has an impact similar to changing basal friction." Fig. 9 shows that there are large differences between calving rates when geometry is kept unchanged and basal sliding is changing. Of course more experiments would give a better picture but should be made in a smaller scale so that the cost of running the HiDEM is not as high. We have rewritten the discussion and the conclusion to express our results in a clearer way:

In the discussion:

[revised manuscript text omitted]

***It is not clear if all the 11 time steps described in Tab. 1 are modeled, or if only a subset of these times are used. Results from t0 , t4 , t6 and t11 are mostly presented, but Fig. 10 also shows results at different time steps.***

The 11 time steps in Table 1 are used to model undercutting. We use all observations/modelled data (front position/runoff) to assess the undercutting for each time step. Thereafter, when using the particle model, we only model four time steps ( $t_0$, $t_4$, $t_6$ and $t_{11}$) for their particularity as the comment in Table 1 shows.

To explain our strategy clearly, we changed a whole paragraph in this section based on the line by line comments which can help readers to understand how each model is separated:

"First, we infer sliding at each time step from surface velocities using an adjoint inverse method implemented in Elmer/Ice with an updated geometry from observations at different time steps. At each iteration, $i$, corresponding to an observed front position, $F_i^{obs}$, the front and the surface are dynamically evolved during the observation time (roughly 11 days) with Elmer/Ice. By the end of the time step, the front has advanced to a new position, $F_{i+1}^{elmer}$. Here we use $i+1$ because this is the position the front would have at $t_{i+1}$ in the absence of calving. Second, given subglacial drainage inferred from modelled surface runoff, a plume model calculates melt rates based on the subglacial discharge for each iteration, which are subsequently applied to the front geometry at subglacial discharge locations. At each iteration, the front geometry takes into account the undercut modelled at the former iteration. Finally, the sliding, geometry and undercut (when applicable) are taken as input to the calving particle model HiDEM for each iteration and a new front, $F_{i+1}^{hidem}$, is computed for four iterations, $t_0, t_4, t_6, t_{11}$, which represent interesting cases (see comments on Table 1). More details about each aspect of the model process are given in the following sections."

***What is the rational for keeping or removing the undercut in one case or another when the ice front advances or retreats (Fig. 4 and p.9)? Some explanations justifying these choices should be added as opposed to presenting the choices made without any justification. I cannot quite figure out why the undercut from the previous profile is not always considered.***

The idea is to take into consideration the observations. One should keep in mind that the undercut estimation is done independently from the HiDEM simulations. The first iteration undercut is estimated from the advanced front, $F_1^{elmer}$ (advanced from $F_0^{obs}$), by projecting daily melt rate during the time period $t_1-t_0$. At the second iteration, $t_2$, we know where the front would be if there had not been any calving between $t_1$ and $t_2$:

$F_2^elmer$, which is the advanced front from the observed position at $t_1$, $F_1^obs$. So we can transfer the whole undercut from previous iteration to $F_2^elmer$ if $F_1^obs$ is situated in front of F1elmer (case a). Otherwise, the undercut would have been fully or partly calved away (case b and c). We then apply the new undercut on this new geometry given the melt rates between $t_1$ and $t_2$.

We changed the text in consequence:

"When the first discharge occurs, the melt rate calculated with the plume model in 2D is summed for the period of time between $t_0$ and $t_1$ and projected to the advanced front $F_1^{elmer}(z = 0)$ (advanced from $F_0^{obs}(z = 0)$) at the location of the subglacial outlets and ice is removed normal to the front. This yields a new position of the front at depth $z$ below sea level called $F_1^{elmer}(z)$. At the second iteration, $t_2$, we know where the front would be if there had not been any calving between $t_1$ and $t_2$: $F_2^{elmer}(z = 0)$, which is the advanced front from the observed position at $t_1$, $F_1^{obs}(z = 0)$. So we can transfer the whole undercut from previous iteration to $F_2^{elmer}(z)$ if $F_1^{obs}(z = 0)$ is situated in front of $F_1^{elmer}(z)$ (see Fig. 4b–c). Otherwise, the undercut would have been fully or partly calved away (see Fig. 4b–c). We then apply the new undercut on this new geometry given the melt rates between $t_1$ and $t_2$.

At time $t_i$, the modelled front position at depth $z$ (advanced by Elmer/Ice from the observed front position at $t_{i-1}$) is $F_i^{elmer}(z)$ and the observed front position is $F_i^{obs}(z = 0)$. We advance this observed front with Elmer/Ice during $\Delta t = t_{i+1} - t_i$ to obtain the front position $F_{i+1}^{elmer}(z = 0)$ at $t_{i+1}$. We want to determine $F_{i+1}^{elmer}(z)$ and depth $z$ given the melt rate calculated between $t_i$ and $t_{i+1}$ and the state of the undercut from the previous front $F_i^{elmer}(z)$ updated by the observed front $F_i^{obs}(z = 0)$."

**Line by line comments:**

*p.1 l.17: "rigorous methods": the problem is not so much about rigorous methods but more about some processes impacting calving that we still don't understand, as well as small scale features (mm long cracks) that cannot be observed*

*and included in models.*

We change to:

"To a large degree, this uncertainty reflects the limited understanding of processes impacting calving from tidewater glaciers and ice shelves, and associated feedbacks with glacier dynamics. In particular, calving occurs by the propagation of fractures, which are not explicitly represented in the continuum models used to simulate ice flow and glacier evolution."

*p.1 l.20: "impacting on submarine melt rate" → "impacting submarine melt rate"*

Done!

*p.2 l.1: "during the summer and the autumn" → "during summer and autumn"*

Done!

*p.2 l.3: "followed by ice-front collapse": not clear*

We change to: "triggering collapse of the ice above"

*p.2 l.12: The problem is actually not so much the representation of calving in models but the processes impacting calving that are not enough understood and therefore cannot be included into models.*

We change to: "In addition to the lack of process understanding, continuum models cannot explicitly model fracture, but must use simple parameterisations such as damage variables or phenomenological calving criteria."

*p.2 l.16: Again here, it is not really coupling but feeding the particule model with appropriate inputs from Elmer/Ice.*

This is true and that is why the word "introduce" is used because it is just an introduction. But we agree that it is misleading. Therefore, we change to:

"In this paper, we use both the capabilities of the continuum model Elmer/Ice and the

discrete element model HiDEM."

*p.2 l.32: "one of the fastest" → "one of the fastest glacier"*

Done!

*p.2 l.33: How much seasonal variation is this glacier experiencing?*

See question below.

*p.3 l.3: How large are the seasonal variations? What is the velocity in winter?*

We add more information: "In 2013, averaged velocities close to the front ranged from 2.2 to 3.8 m d$^{-1}$ in the summer and fell to 2 m d$^{-1}$ directly after the melt season. In 2014, however, they stayed relatively high (around 4 m d$^{-1}$) throughout the summer and progressively fell to 3 m d$^{-1}$ in the winter."

*p.3 Fig.1: Consider adding Kongsfjorden on the figure. Calving front position for 16 Octobre 2013 is not visible, consider changing it.*

Done!

*p.3 l.10 "by (Nahrgang et al., 2014)" → "by Nahrgang et al. (2014)"*

Done!

*p.3 l.11-15: Past tense should be used to describe measurements made in 2013.*

Done!

*p.3 l.11-15: How representative of the seasonal cycle are these values?*

These values are only valid for the summer season.

*p.4 Fig.2: There is no feedback and therefore no coupling shown on this figure. Outputs from one model are used as parameters/inputs for the next model.*

We therefore refer to it as a one-way coupling approach. We change the first sentence

of the section though to:

"We use surface velocity and frontal position data described above to test the effects of sliding and undercutting on calving using different models in a global approach."

*p.4 l.7: What about the other observations (geometry, ice temperature and viscosity)? Where do they come from?*

Geometry was described in section 3.3 but we agree that all observations should be presented together so we moved it and changed the title of the section to "3.1. Observed geometry, surface velocities and front positions". There is no observation of ice temperature or viscosity and more details on the Elmer/Ice model description is given in Vallot et al. (2017), cited in section 3.3. To be clear, we added this information: "More details on the model (viscosity, ice temperature, inversion time-steps, etc.) are given in Vallot et al. (2017)."

*p.5 l.6: Is the sliding inverted just at the beginning of the simulation or recomputed for each time step? In this case how is the change in the glacier geometry computed (or maybe observations are used)?*

The sliding is inverted at each time step given the observations (velocity, topography, front position). Thereafter the surface elevation is relaxed using Eq. 2 in a transient simulation. This new surface is then used in the next iteration. This is done independently of the HiDEM and is only using front position observation. This is what should change in a full coupling. To avoid confusion, we add "at each time step".

*p.5 l.9: How are the front and surface evolved? Are they run from the previous iteration and therefore the 12 time steps are run with the model? In this case, why only show results for 6 cases and not the entire melt season? If not, how are the front and surface evolved? Also, modeling ice front changes in ice flow model is not an easy task and is currently the subject of active research. How is the front evolved with the Elmer/Ice model? There is no reference or explanation of*

[Figure]

*how the ice front migrates and no Elmer/Ice paper describing such an evolution to my knowledge. This has to be better explained.*

The front and surface are evolved in a transient simulation using Elmer/Ice, as explained in the following section (3.3). We add more information on the front evolution:

"The front is advanced by imposing a Lagrangian scheme over a distance equal to the ice velocity multiplied by the time step. We do not account for the melting during the advance because we only have observations at the beginning and the end of each timespan. Instead, we lump frontal melting by applying an undercut after the advance as explained hereafter."

Since it is not a full coupling, the results from the HiDEM are not used in Elmer/Ice where we only use front position observation. This section was intended to be more a presentation of the modelling concept with more explanation to follow. Is it misleading? We add a sentence referring to the next paragraphs for more details:

" More details about each aspect of the model process are given in the following sections."

Also, the sentence about the front and surface evolution is misleading since it seems that the front evolves from one observation to another, which is not the case. So we change it to:

"the front and the surface are dynamically evolved during the observation time"

*p.6 l.1: "coupling": same as above*

Here we explain that we use "one-way coupling" with input from a model is output from another but not vice-versa.

*p.6 l.1: If the front position is not used as inputs for the Elmer/Ice initial front position, what is used then?*

We use the observed frontal position for Elmer/Ice simulations. We add this information: "In Elmer/Ice, we use the observed frontal positions."

***p.6 Eq. 1: Consider using vectors. Also u is used both here for the velocity, and later (e.g. Tab. 2) for the undercutting. Change one or the other.***

Thank you for this suggestion! We chose to change the velocity to v and we change to vectors.

***p.6 l.8: Again here, is the friction optimized at each time step or just at the beginning of the simulation?***

The friction is optimised at each time step for which we have observations. We add this information: "The basal friction coefficient, $\beta$, is optimized at each time step to best reproduce observed velocity distribution at the surface of the glacier as described in Vallot et al. (2017)."

***p.6 l.9: "the self-adjointness" → "a self-adjoint algorithm"***

Done!

***p.6 l.10 and l.11: Consider adding older references that first used such methods.***

We add references:

"This is done by using a self-adjoint algorithm of the Stokes equations for an inversion (e.g. Morlighem et al., 2010; Goldberg and Sergienko, 2011; Gillet-Chaulet et al., 2012) and implemented in Elmer/Ice (Gagliardini et al., 2013)."

***p.6 l.13: This paragraph could be put in the data section (section 3.1) to improve consistency.***

Done!

***p.6 l.20: "The front position is also able to advance": How is it able to advance? See point above***

As described above, we add the information: "The front is advanced by imposing a

Lagrangian scheme over a distance equal to the ice velocity multiplied by the time step."

**p.6 l.20: "$F^o_i bs(0)$": I would imagine that observations show the front position on the surface of the glacier and not at sea level.**

We assume that the front is vertical above sea level. We add this information: "We assume that the front is vertical above the sea level so that the observed front position (at the surface of the glacier) is the same at sea level."

**p.6 l.20-21: There are several front positions observed and computed. The authors should start by listing all the front position computed ($F^{elmer}$, $F^H iDEM$, ...) and explaining where they come from. That might be something to add on Fig.2.**

Thank you for the suggestion! We added this information in the section 3.2:

"First, we infer sliding at each time step from surface velocities using an adjoint inverse method implemented in Elmer/Ice with an updated geometry from observations at different time steps. At each iteration, $i$, corresponding to an observed front position, $F^{obs}_i$, the front and the surface are dynamically evolved during the observation time (roughly 11 days) with Elmer/Ice. By the end of the time step, the front has advanced to a new position, $F^{elmer}_{i+1}$. Here we use $i + 1$ because this is the position the front would have at $t_{i+1}$ in the absence of calving. Second, given subglacial drainage inferred from modelled surface runoff, a plume model calculates melt rates based on the subglacial discharge for each iteration, which are subsequently applied to the front geometry at subglacial discharge locations. At each iteration, the front geometry takes into account the undercut modelled at the former iteration. Finally, the sliding, geometry and undercut (when applicable) are taken as input to the calving particle model HiDEM for each iteration and a new front, $F^{hidem}_{i+1}$, is computed for four iterations, $t_0, t_4, t_6, t_{11}$, which represent interesting cases (see comments on Table 1). More details about each aspect of the model process are given in the following sections."

$F^{obs}$ and $F^{elmer}$ are already in Fig. 2.

***p.6 section 3.3: What is the resolution (horizontal and vertical) of the model, especially close to the ice front? What are the time steps used for the continuum model?***

We add this information:

"We use an unstructured mesh, with spatial repartition of elements based on the mean observed surface velocities in the horizontal plane (roughly 30 m resolution close to the front). Vertically, the 2D mesh is extruded with ten levels (roughly 10 m resolution close to the front)."

Later in the section (after the surface equation), we add:

"We use a time step of 1 day."

***p.7 l.8: convention for the reference (twice)***

Corrected!

***p.7 l.29: "five kilometers" → "five kilometers away"***

Done!

***p.8 l.1: How long does it take to reach a steady-state?***

We have clarified this in the text, p.8 l.1 now reads: "The model is spun-up for 1000 model seconds until the turbulent kinetic energy in the region of the plume reaches a steady state..."

***p.8 l.4-7: So my understanding is that the discharge varies but not the ocean conditions. Ocean conditions are reported quite accurately on p.3, so why not use these conditions instead of uniform ambient ocean properties? Also, in all these cases, the ice front is assumed to be vertical, why not try cases with pre-existing undercutting? I understand that it might not be possible to test all these***

***cases, but at least assessing the uncertainty caused by such assumptions would be important.***

The model uses temperature and salinity profiles collected from Kongsfjorden, as described on p7. l.26–30. They are uniform in the sense that the same conditions were used in the different discharges tested, we have rewritten to clarify this. Edited p.8. l.5 to "Instead, representative cases $M_d$ using the ambient ocean properties described above and discharges d of 1, 10, 50 and 100 $m^3 s^{-1}$ were tested and the melt rate profiles for intermediate discharges were linearly interpolated from these cases."

The reasons for and implications of not varying the ocean properties and the ice front angle are discussed in Section 5.1. However, it is important to give the full picture and we have replaced the paragraph on ocean variability in the discussion (p.19 l.3-12) with the paragraph below:

"By using ambient temperature and salinity profiles that do not vary in time, we neglect the inter- and intra-annual variability in Kongsfjorden. This variability can affect the calculated melt rate in two ways: i) the three-equation melt parameterisation explicitly includes the temperature and salinity at the ice-face, and ii) the ambient stratification affects the vertical velocity and neutral buoyancy height of the plume. The direct effect of changes in temperature and salinity on the melt equations are well tested. Past studies using uniform ambient temperature and salinity conditions have found a linear relationship between increases in ambient fjord temperatures and melt rates, with the slope of the relationship dependent upon the discharge volume (Holland et al., 2008b; Jenkins, 2011; Xu et al., 2013). Salinity, on the other hand, has been shown to have a negligible effect on melt rates (Holland et al., 2008a). However, with a non-uniform ambient temperature and salinity, the effects of changes in the stratification on the plume vertical velocity and neutral buoyancy are much more complex. The stratification in Kongsfjorden is a multi-layer system, with little or no direct relationship between changes in different layers (Cottier et al., 2005). Therefore, testing cases by uniformly increasing or decreasing the salinity would not be informative for understanding the true effects of

inter- and intra-annual variability. The high-computational expense of the plume model used here means that it is not yet feasible to run the model on the timescales necessary to understand this variability, nor to run sufficient representative profiles to provide a useful understanding of the response. Previous work has suggested that intra-annual changes in the ambient stratification are small enough that plumes are relatively insensitive to these changes (Slater et al., 2017) and that plume models forced with variations in runoff and a constant ambient stratification can qualitatively reproduce observations (Stevens et al., 2016). For these reasons, we highlight this as a limitation of the current implementation, and suggest that this should be addressed in future investigations of plume behaviour. A model based upon one-dimensional plume theory (e.g. Jenkins, 2011; Carroll et al., 2015; Slater et al., 2016) would be less computationally expensive and may allow some of these limitations to be addressed. However, such a model would not capture the strong surface currents driven by the plume which are important for the terminus morphology studied here."

*p.9 l.6-10: What is the rational for keeping or removing the undercut in one case or another? Some explanations justifying these choices should be added.*

See answer to major comment.

*p.9 l.19: How many broken beams are added and how was this number chosen? What is the impact of increasing or reducing this number on the results? Also does the number of broken beams increase during the melt season as the ice gets more damaged?*

This must be some kind of misunderstanding on how the model is constructed - we do not add broken beams at any point. What the referee probably refers to is the small fraction of broken beams at the beginning of the simulation. As long as this fraction is small and broken beams are spatially uncorrelated it has only a minor influence on calving. It is a good suggestion by the referee to increase this fraction to mimic melt and to investigate how that would affect calving. However, in our opinion it would not

be useful to add even more results to this paper and yet another calving variable in a model which already has a lot.

***p.9 l.30: How long is the HiDEM model run for at each time step? And how long does it take to run it?***

We add this information: " The model run for 100 s, which takes two days of simulation physical time."

***p.10 l.2: What kind of instabilities are developing and why?***

We add more information to make this statement clearer:

"HiDEM reads a file with surface and bed coordinates on a grid and a file with surface and basal ice (to take into account the undercut) coordinates. When simulating with an undercut at a discharge location and in order to avoid complication in the HiDEM (position of the basal ice), we remove particles below the maximum melt (no ice foot)."

***p.10 l.7-11: What is the rational for decreasing the friction? How is the choice of friction impacting your results?***

There is a clear separation of timescales between the velocities of sliding ($\sim$m/day) and calving ice ($\sim$m/sec). This means that as long as sliding is slow enough to be negligible during a single calving event, we can change it without much effect on any single calving event. As an approximation we can assume that fast processes are at equilibrium when we consider slower timescales. However, a rescaling speeds up the frequency of calving, and we can thus 'speed up', within reason, the few minutes of HiDEM simulation to effectively model calving which would otherwise take tens of hours or days, and thus be practically impossible to simulate with HiDEM. By applying scaling, the calving events modelled during the simulation of HiDEM (few minutes) correspond to the sum of calving events that would happen during the time scale of sliding. The scaling factor that we use is the same for the whole domain and for all simulations. We use a friction scaling factor for $\beta$ equal to $10^{-2}$ (or sliding velocity scaled up by $10^{2}$),

and simulations run until calving stops and a new quasi-static equilibrium is reached. This is now better explained in the text:

"There is a clear separation of timescales between the velocities of sliding ($\sim$m day$^{-1}$) and calving ice ($\sim$m sec$^{-1}$). This means that as long as sliding is slow enough to be negligible during a single calving event, we can change it without much effect on any single calving event. As an approximation we can assume that fast processes are at equilibrium when we consider slower timescales. However, a rescaling speeds up the frequency of calving, and we can thus 'speed up', within reason, the few minutes of HiDEM simulation to effectively model calving which would otherwise take tens of hours or days, and thus be practically impossible to simulate with HiDEM. By applying scaling, the calving events modelled during the simulation of HiDEM (few minutes) correspond to the sum of calving events that would happen during the time scale of sliding. The scaling factor that we use is the same for the whole domain and for all simulations. We use a friction scaling factor for $\beta$ equal to $10^{-2}$ (or sliding velocity scaled up by $10^2$), and simulations run until calving stops and a new quasi-static equilibrium is reached."

***p.10 l.19: It should be mentioned that this is volumetric ablation rate (same for volumetric calving rate in the rest of the paper). Many people use calving/ablation rate as changes per unit area (in m/yr), which can be confusing.***

We add "mean volumetric" in front of ablation rate and calving rate in the whole paper.

***p.10 l.20: Integrals over Gamma usually refer to contour intervals and not surface integrals, using $S$ or $\Sigma$ instead would be more consistent with literature.***

Thank you for the suggestion. We change $\Gamma$ for $\Sigma$.

***p.10 Eq.5: What is $z\Gamma_w$ ?***

This was meant to be the vertical dimension of (now) $\Sigma$. We change it to $z \in \Sigma$.

***p.10 l.28: "parameterisations" $\rightarrow$ "parameters"***

Done!

**p.10 l.30: u was already used for velocity (see above)**

We change the velocity for v and kept u for undercut (see above).

**p.10 l.30: Only 4 time steps are mentioned here. What happens to the other ones, are they just excluded? In this case, what is used for the prior undercut?**

Undercuts are computed from for each timestep for which there are observations, so independently from theand HiDEM simulations were conducted for four time steps ($t_0$, $t_4$, $t_6$, and $t_{11}$). This is now clearly expressed in the text and Tables.

**p.11 l.1: Only a subset of $(i, j) \in [0, 4, 6, 11]$ is covered, not accurate.**

We remove this.

**p.11 l.6: configuration $C_k$ is not defined and not used anywhere else, should be consistent with the rest of the paper**

That is right. We changed the names of the configurations at a later stage but the table had not been updated. This is now done!

**p.11 Tab. 2: Configuration is here a function of time ($t_i$) as opposed to geometry ($g_i$) in the rest of the paper**

See above.

**p.12 Fig.6 caption: "data gaps corresponds" → "data gaps correspond"**

Done!

**p.13 l.Tab.3: Discharged should be provided in m3/s to be consistent with the rest of the text. No data between $t_{10}$ and $t_{11}$ , this should be added even if the values are just zero. Also, how are the melting rates for each case computed based on Fig.7? Is an interpolation between the four cases been performed? Or something else?**

In Table 3, we wanted to show the total volume discharged during the time period, hence a volume in m3. However, if it is irrelevant, we could provide an averaged discharge in m3/s. Wee added $t_{10}$ to $t_{11}$. Yes, melting rates are interpolated from the four cases and this is mentioned in section 3.5:

"discharges $d$ of 1, 10, 50 and 100 $\mathrm{m^3\,s^{-1}}$ were tested and the melt rate profiles for intermediate discharges were linearly interpolated from these cases"

**p.13 Fig.7: How different are the results if there is undercut introduced in the geometry?**

We discuss the impacts of this in Section 5.1. Slater et al. (2017) show that undercuts only have a weak effect on plume dynamics, and therefore our projection of the melt rates onto the terminus is a reasonable way to address this, based on research to date. This is an area which requires further investigation with high-resolution plume models; however, as this is not the primary focus of this paper we suggest that once other studies have addressed this, their results, models and methodologies can be incorporated into future development of the approach we present here.

**P.14 Fig.8: It is the only time in the paper, where results from times other than t0 , t4 , t6 and t11 are presented. Are the other time steps computed? And what is the rational to only present some ice front positions here?**

The undercut estimation from melt rates is independent from the HiDEM simulations. We use daily runoff and observed front positions for it. So in order to get the undercut for $t_6$, we need to know the undercut state from previous steps. We do not present the front positions after $t_6$ since we do not use them for the HiDEM simulations. For $t_{11}$, we consider the front vertical (the remained undercut from last iteration with melt being calved away).

**p.14 Fig.8: If z is the height above sea level, Fig.8 b and c are for z = -3m and z = -42m, and the stars in Fig.8 d and e indicate the plan view elevation, why are**

*the start not aligned at the same height on Fig.8 d and e? Also it might be more clear to use "Elevation from sea level" or something similar instead of "Distance to the bed" on Fig.8 d and e as this is what is used in the rest of the figure.*

The stars are not aligned because the sea level is actually not at the same position compared to bed elevation for each iteration. We chose to use "distance to bed" instead of "elevation from sea level" because the plume starts at the bed elevation and we wanted to compare from there.

*p.14 l.11: What do the authors mean by "smoother"?*

We mean the undercut is not as abrupt as ND: We change the sentence to:

"In the first 50 m from the surface, the undercut at the SD is not as abrupt as at the ND and is also smaller"

*p.14 l.12: "stretching" → "stretches"*

Done!

*p.15 l.2: Why not do it more often? How long does it take to run the model?*

The runoff is available everyday and thus the discharge. Undercutting is therefore applied everyday based on the observed front position. Unfortunately, we only have observed front every 11 days and that is why we do not do it more often. We add this information:

"One should keep in mind that our modelling approach neglects the change of the front during the period of interest between two observations of frontal positions (11 days for most cases)."

*p.15 Tab.4: "modelled" → "estimated": the melt is estimated from observations, not modeled. What is $\dot{a}_m^o bs$ ? Also add zero values where appropriate instead of leaving empty spaces.*

This was a mistake, this should be $\dot{a}_m$.

*p.16 Fig.9: $\dot{a}_c = \dot{a}_{c,u} - \dot{a}_{c,L}$ on p.15 (minus not plus), so it's not clear what is shown on this figure.*

$\dot{a}_{c,L}$ is actually negative so what is shown on the figure is $-\dot{a}_{c,L}$. We change the figure accordingly.

*p.19 l.5: "Due to this, there is some uncertainty" → "There is therefore some uncertainty"*

Done!

*p.19 l.7: Add references (or something else) to justify this statement that is not supported.*

We have rewritten and added references to support this statement:

"Previous work has suggested that intra-annual changes in the ambient stratification are small enough that plumes are relatively insensitive to these changes (Slater et al., 2017) and that plume models forced with variations in runoff and a constant ambient stratification can qualitatively reproduce observations (Stevens et al., 2016)."

*p.19 l.24: Is that what is observed by the authors?*

Yes.

*p.20 l.1: "no retreat at all": this is not supported by the model results, there is some retreat in the southern part of the domain.*

We meant no retreat at all at SD. We add this information.

*p.20 l.9-10: I don't think that the experiments made and results support such a conclusion, the role of sliding and geometry cannot be clearly separated.*

See answer to major comment and new rewriting of the section.

***p.20 l.19: "the velocity higher" → "the higher velocity"***

Done!

***p.20 l.19: "seem" → "seems"***

Done!

***p.20 l.30: "reproduces" → "reproduced"***

Done!

***p.21 l.12: "in 2D" → "with a simplified 2D geometry"***

Done!

***p.21 l.14: How do the melt rates compare to previous results?***

There are no previously published frontal melt rates for this glacier (modelled or observed), so no comparison is possible.

***p.21 l.23: What is referred to as the calving model?***

We mean the discrete particle model. The text has been changed to make this clear.

***p.22 l.5: "would be implemented" → "were implemented"***

Done!

---

## Author Response (AR2)

**Answer to referee and editor**

**Effects of undercutting and sliding on calving: a global approach applied to Kronebreen, Svalbard**

Dorothée Vallot[1], Jan Åström[2], Thomas Zwinger[2], Rickard Pettersson[1], Alistair Everett[3], Douglas I. Benn[4], Adrian Luckman[5,6], Ward J. J. van Pelt[1], Faezeh Nick[7], and Jack Kohler[3]

[1]Department of Earth Sciences, Uppsala University, Sweden
[2]CSC - IT Center for Science, Espoo, Finland
[3]Norwegian Polar Institute, Fram Centre, N-9296 Tromsø, Norway
[4]School of Geography and Sustainable Development, University of St Andrews, St Andrews, Scotland, UK
[5]Department of Geography, Swansea University, UK
[6]Department of Arctic Geophysics, UNIS, The University Center in Svalbard, Longyearbyen, Norway
[7]Arctic Geology Department, University Centre in Svalbard, Norway

*Correspondence to:* Dorothée Vallot

We first want to thank the referee and the editor for the constructive comments. Our answers to the questions are as follows.

**Suggestion for revision**

*I have now read this new version of the paper which has indubitably been improved after this first round of reviews. I nevertheless still think that there is room for improvements. After this second careful reading, I really think that the paper*

5   *should be presented differently. As I now understand the work, I would emphasize that the objective is a validation of the HiDEM model against a datatset, the 5 other models being here "only" to construct the required input fields for this validation, as clearly depicted in Fig. 2 (where HiDEM is the only model which doesn't appear directly in the flow chart). This might require some substantial rewriting at different places of the manuscript to make this clearer. For example, at the bottom of p. 5, it should be emphasized that you are using 5 models to build the appropriate input fields for the HiDEM*

10   *model, in order to compare the modelled calving to the observed one.*

Our aim in this paper was to conduct a comparison of observed and modelled calving front positions, not simply to validate HiDEM. We have made clearer statements of our aims in the Abstract and Methods.

In the abstract, we add:

"We demonstrate the feasibility of reproducing the observed calving retreat at the front of Kronebreen, a tidewater glacier in

15   Svalbard, during a melt season by using the output from the first five models as input to HiDEM."

In section 3.2, we add:

"In this paper, we use the output of five different models as input for the discrete particle model, HiDEM, in order to compare the modelled calving front to observations for different configurations of sliding, geometry and undercutting."

We have also changed "one-way coupling" to "offline coupling" to make things clearer.

**Other remarks**

*p. 1, l. 8: with different discharges.*

Done.

*p. 2, l. 23: the previous sentences are general regarding the comparison between continuous and discrete models, and should be clearly separated with the end of the paragraph which is only related to the present study.*

We added a new paragraph.

*Table 1: you should mention in this table at which observational time is run the discrete model (as one understands later, it is not run at each observational times).*

The observational times at which the discrete model is run have been identified by gray rows in the table and in the caption:
"The HiDEM model is run for observational times $t_0$, $t_4$, $t_6$ and $t_{11}$ indicated by the gray color."

*p. 6, l. 5: the time step used by Elmer/Ice should be given here. Also, a clear distinction between model time step (1 day for Elmer/Ice as it is mentioned p. 7) and observation time (and time step, as mentioned in Table 1) should be made all along the manuscript. For example, l. 9, to which "iteration" are you refereeing to? The Elmer/Ice time step iterations or the observational dates?*

To make the distinction clearer, we use observation times for $t_i$, iteration for $i$ corresponding to an acquisition time, observation interval for $\Delta t_i$ and time step for Elmer/Ice or HiDEM transient time steps all along the manuscript.

Caption of Table 1: "Observation times of velocity acquisitions, $t_i$, associated dates and time interval between two observations $(\Delta t_i)$."

p. 6, l. 3-13: "First, we infer sliding at each time step from surface velocities using an adjoint inverse method implemented in Elmer/Ice with an updated geometry from observations for the different velocity acquisitions. At each iteration, $i$, corresponding to an observed front position, $F_i^{obs}$, the front and the surface are dynamically evolved during the observation time interval (roughly $11\,\mathrm{days}$) with Elmer/Ice with a time step of $1\,\mathrm{day}$. By the end of the observation interval, the front has advanced to a new position, $F_{i+1}^{elmer}$. Here we use $i+1$ because this is the position the front would have at $t_{i+1}$ in the absence of calving. Second, given subglacial drainage inferred from modelled surface runoff, a plume model calculates melt rates based on the subglacial discharge for each iteration, which are subsequently applied to the front geometry at subglacial discharge locations. At each iteration, the front geometry takes into account the undercut modelled at the former iteration. Finally, the sliding, geometry and undercut (when applicable) are taken as input to the calving particle model HiDEM for each iteration and a new front, $F_{i+1}^{hidem}$, is computed for four iterations, $i = \{0, 4, 6, 11\}$, which represent interesting cases (see comments on Table 1). More details about each aspect of the model process are given in the following sections."

*p. 6, l. 11: then, following my main remarks, why just using 4 observational dates for the validation of HiDEM and not extend it to the whole observational series you have?*

This could have been possible but would have required more computer resource. We therefore chose to focus on the more interesting cases.

*p. 6, l. 16: the output of HiDEM,*

Done.

*p. 7, l. 13: More details on the Elmer/Ice modeling*

Done.

*p. 7, l. 15: The temporal evolution of the glacier is not only given by equation (2) but also the coupling with the Stokes solution which gives the velocity as an input for the evolution of the glacier geometry? Your are never mentioning the forward model used to compute the ice velocity. Regarding the velocity calculation, how is basal sliding evolved during the 11 time steps? Especially where it has not been inverted in the case of an advancing front? Also, at other location, is it interpolated between i and i+1 observational dates?*

After describing Eq. 2, we add:

"We use a time step of $1 \, \mathrm{day}$ during the interval of time between two acquisitions. Eq. 2 is solved alongside the Stokes equation, coupled to the latter by the velocities. The basal sliding is not evolved and stays equal to the result of the inversion. When the front is advanced, the mesh is stretched to match the new front position. No new element or node is created and the basal sliding values are extrapolated towards the new front. The new surface is in fact only used as an input for the next iteration. There is no interpolation of the basal sliding between two observational dates."

*p. 8, l. 13: modelled on a 100x 100 $m^2$ grid*

Done.

*p. 9, l. 16: advanced front from Elmer/Ice and melt rates from the plume model to estimate*

Done.

*p. 9, l. 17: You might specify the daily undercutting is subtracted to the front surface only every observational time step?*

We add:

"At each iteration, $i$, the sum of the daily undercutting during the observation interval is subtracted from the front."

*Fig. 4: The black undercut curve is confusing as only the red and grey ones are discussed in the legend. What does it stand for? Also, the side views show ice foot, and a new colour l. should be added to indicate that, as said for the discrete model, ice foot are not accounted for. You should then illustrate on this figure what is the simplified front geometry given as an input to HiDEM.*

The legend has been changed to include the black line, the HiDEM input has been added and the caption has been changed accordingly:

"Three cases of undercut $i + 1$ at $t_{i+1}$ (black line) depending on former undercut $i$ at $t_i$ (gray line) at $z$ relative to $F_i^{obs}(z = 0)$ (black line with circles) in plan view (left) and side view (right). The red star represents the discharge location. On the side view, the dashed line represents the simplified undercut geometry where the ice foot has been removed, which is given as input to the HiDEM. **(a)** $F_i^{obs}(z = 0)$ is behind $F_{i+1}^{elmer}(z = 0)$ and in front of $F_i^{elmer}(z)$. The undercut from $F_i^{elmer}(z)$ is translated to $F_{i+1}^{elmer}(z)$ (gray line) and the new undercut is superposed (red line). **(b)** $F_i^{obs}(z = 0)$ is in front of $F_i^{elmer}(z)$. The remnant from $F_i^{elmer}(z)$ (what is behind $F_i^{obs}(z = 0)$) is translated to $F_{i+1}^{elmer}(z)$ (gray line) and the new undercut is superposed (red line). **(c)** $F_i^{obs}(z = 0)$ is behind $F_i^{elmer}(z)$. The undercut from $F_i^{elmer}(z)$ is ignored and the undercut created at $t_{i+1}$ is the only one (red line)."

*p. 11, l. 9: Melt above the sea surface (?)*

"Frontal melt above the sea surface"

*p. 11, l. 11: 10x10 m$^2$ grid*

Done.

*p. 11, l. 12: is this new front modeled by HiDEM at each observational dates or only at the four selected ones? This should be mentioned.*

We add:

"for the four selected iterations ($i = \{0, 4, 6, 11\}$)."

*p. 11, l. 29: two days of computing time. (?)*

Done.

*p. 11, l. 30: with HiDEM thereafter. When? Every observational times or only for $t_0$, $t_2$, $t_4$ and $t_{11}$?*

We add:

"for the selected iterations."

*p. 12, l. 6: the no ice foot geometry should be indicated in Fig. 4.*

The ice foot geometry has been added to Fig. 4 and in the text, we add:

"as shown by the dashed line in Fig. 4"

*p. 12, l. 13: a rescaling of what? The sliding coefficient? Then this sentence seems to say exactly the opposite of the previous one? It has no influence but it speeds up...*

We change the text from:

"This means that as long as sliding is slow enough to be negligible during a single calving event, we can change it without much effect on any single calving event. As an approximation we can assume that fast processes are at equilibrium when we consider slower timescales",

to:

"This gives us the opportunity to rescale friction so that we can more effectively simulate calving: even if we scale down friction by e.g. two orders of magnitude and increase sliding accordingly to $\sim 100\,\mathrm{m\,day}^{-1}$, there is still negligible sliding during calving events which last tens of seconds or perhaps a minute."

*p. 12, l. 21: the next observed configuration.*

Done.

*Caption Fig. 5: Basal friction coefficient obtained from inverse modelling*

Done.

*Caption Fig. 9: there is no hashed red box. "The mean distance differences between the modelled and the observed front positive and negative" -> "The mean distance differences between the modelled and the observed front positions"*

Done.

*Fig. 10: You should indicate which model curves should be compared to observations? Which curves are expected to be superimposed?*

All colored curves are expected to be compared to observations.

*p. 22, l. 16: Remarkably may be a bit exaggerated in light of Fig. 10?*

We remove "remarkably"

*p. 22, l. 25: It is related to the rules used to estimate calving in HIDEM. An other rule could be that the block should not be grounded anymore?*

This is s good idea and could be used in future work.

**Additional minor points by editor A. Vieli**

*p. 4 Fig. 1: in figure (a) I really struggle to see the 'four frontal positions' mentioned in the caption. Make clearer in caption of Fig. 1, l. 1: to me the bare rock seems rather 'brown' than 'red'.*

The front positions have been moved to Fig. 1b. Red has been replaced by brown.

*p. 5 Fig. 5: as mentioned above maybe integrate the HiDEM model in this schematic.*

The input to HiDEM have been highlighted with a particular legend. We think that adding HiDEM would make the schematic more difficult to read.

*p. 6 l. 3: I assume you refer to 'sliding speed' or 'sliding coefficient' here, 'sliding' alone seem rather vague.*

We change to "sliding velocity".

*p. 6 l. 4: should it not say 'iteration $i$ to $i+1$' instead of just 'iteration $i$'.*

Each iteration corresponds to an acquisition time.

*p. 6 l. 10: again again 'the sliding. . . are taken', . . . 'sliding what?', velocity. . . .?*

We change to "sliding velocity".

*p. 7 l. 18: replace 'in reaction to' with 'for a' as you would only expect a reaction if you have a CHANGE in accumulation.*

Done.

*p. 7, l. 21: I would replace 'sea level' by 'water line'*

Done.

*p. 7 l. 22: 'We call. . . '*

Done.

*p. 8 l. 2: '. . . SUBMARINE melting during. . . '*

Done.

[revised manuscript text omitted]